# AnomSeer: Reinforcing Multimodal LLMs to Reason for Time-Series Anomaly Detection

Junru Zhang[1]  Lang Feng[2]  Haoran Shi[2]  Xu Guo[2]  Han Yu[2]  Yabo Dong[1]  Duanqing Xu[1]

## Abstract

Time-series anomaly detection (TSAD) with multimodal large language models (MLLMs) is an emerging area, yet a persistent challenge remains: MLLMs rely on coarse time-series heuristics but struggle with multi-dimensional, detailed reasoning, which is vital for understanding complex time-series data. We present ANOMSEER to address this by reinforcing the model to ground its reasoning in precise, structural details of time series, unifying anomaly classification, localization, and explanation. At its core, an expert chain-of-thought trace is generated to provide verifiable, fine-grained reasoning from classical analyses (e.g., statistical measures, frequency transforms). Building on this, we propose a novel time-series grounded policy optimization (TimerPO) that incorporates two additional components beyond standard reinforcement learning: a time-series grounded advantage based on optimal transport and an orthogonal projection to ensure this auxiliary granular signal does not interfere with the primary detection objective. Across diverse anomaly scenarios, ANOMSEER, with Qwen2.5-VL-3B/7B-Instruct, outperforms larger commercial baselines in classification and localization accuracy, particularly on point- and frequency-driven exceptions. Moreover, it produces plausible reasoning traces that support its conclusions.

## 1. Introduction

Recent advances in large language models (LLMs) have opened new opportunities for time-series anomaly detection (TSAD) (Xu et al., 2021). Building on this progress,

we focus on a practical yet underexplored setting, *time-series reasoning for anomalies* (Yang et al., 2025; Kong et al., 2026), where the goal goes beyond flagging abnormal segments: models must also produce coherent, linguistically grounded explanations. Emerging studies (Zhou & Yu, 2024; Xu et al., 2025; He et al., 2025) have revealed that LLMs exhibit stronger zero-shot robustness when reasoning over visual renderings of time series (e.g., line plots) rather than raw numeric sequences. This advantage arises from human-like pattern perception and greater token efficiency enabled by compact, semantically rich images (He et al., 2025; Liu et al., 2024). These insights naturally motivate multimodal LLMs (MLLMs) as the backbone for advancing TSAD in a *reasoning-centric* manner, i.e., detecting, attributing, and justifying anomalies with structured natural language grounded in visual cues.

Despite these strengths, MLLMs fundamentally lack built-in time-series priors, and their reasoning often resorts to coarse time-series heuristics and struggles with detailed time-series analysis (Figure 1 (Middle)), thereby leading to suboptimal performance. While reinforcement learning (RL) (Sutton & Barto, 2018) has proven more effective than supervised fine-tuning (SFT) (Zhang et al., 2025b; Liu et al., 2025b; Luo et al., 2025; Tan et al., 2025) at incentivizing the emergent reasoning of LLMs in other domains (Guo et al., 2025; Wei et al., 2025; Feng et al., 2025), its reliance on globally verifiable rule-based goals may be ill-suited for the model to capture subtle, fine-grained time-series patterns. Consequently, even well-trained MLLMs may only excel at salient, out-of-range anomalies yet struggle to articulate nuanced shifts (e.g., small trend drifts) with faithful textual evidence. This discrepancy raises a central question for MLLMs in TSAD:

> *Can we incentivize MLLMs to ground their time-series reasoning in fine-grained, multi-dimensional evidence, ensuring faithful and verifiable anomaly interpretations?*

To address this challenge, we propose ANOMSEER[1], a novel time-series MLLM post-training approach that not only detects anomalies but also produces structured, evidence-based

---

[1]Zhejiang University [2]Nanyang Technological University. Correspondence to: Xu Guo <xu.guo@ntu.edu.sg>, Yabo Dong <dongyb@zju.edu.cn>.

*Proceedings of the 43rd International Conference on Machine Learning*, Seoul, South Korea. PMLR 306, 2026. Copyright 2026 by the author(s).

[1]https://github.com/jrzhang33/AnomSeer.

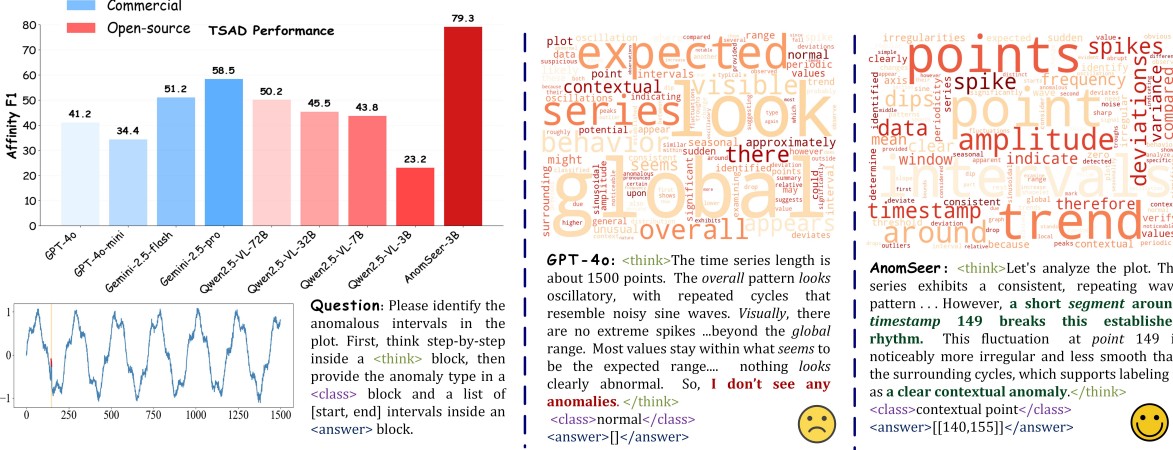

*Figure 1.* Comparison of model performance and time-series reasoning quality. **Left**: Affinity F1 (%) of different models on TSAD benchmarks. **Middle**: GPT-4o results, including word frequency distributions in reasoning (top) and its coarse-grained answer (bottom). **Right**: ANOMSEER results, including word frequency distributions in reasoning (top) and its fine-grained answer (bottom).

explanations to support its decisions. Our core idea is to fuse the analytical rigor of classical numerical TSAD with the holistic visual intuition of MLLMs through two components: *(i) expert chain-of-thought (ExpCoT)* trace, which encodes structured reasoning inspired by classical TSAD workflows, and *(ii) time-series grounded policy optimization (TimerPO)*, a novel temporal-aware RL algorithm that softly aligns the model's reasoning with ExpCoT trajectories. Instead of merely correcting outputs, ANOMSEER utilizes the analytical rigor of traditional TSAD methods, such as residual inspection (Hyndman & Athanasopoulos, 2018) and wavelet-based drift detection (Thill et al., 2017), and embeds it into the MLLM's learning process. TimerPO operationalizes this integration by measuring the semantic deviation from an ExpCoT using optimal transport (Caffarelli & McCann, 2010; Bonneel et al., 2011) and transforms this distance into a refinement advantage signal. This signal is then orthogonally projected, ensuring it acts as non-interfering auxiliary guidance of the main RL objective. Consequently, TimerPO enhances the model's fine-grained temporal-aware reasoning capabilities (Figure 1 (Right)) without perturbing its global visual understanding or the primary optimization objective. We summarize our key contributions as follows:

- We explore a pivotal challenge hindering the effectiveness of MLLMs for TSAD: the tendency of MLLMs to rely on coarse visual "eyeballing" rather than engaging in fine-grained numerical reasoning. We introduce ANOMSEER, a novel approach that bridges this gap by transferring classical, detailed TSAD priors into the time-series reasoning process of MLLMs during training.

- We propose TimerPO, a new RL algorithm designed for time-series reasoning in TSAD. TimerPO guides fine-grained, numerical time-series knowledge into the

model's reasoning. It leverages optimal transport to create auxiliary advantage signals and applies them as non-interfering corrective guidance for RL training via orthogonal projection.

- Extensive experiments across diverse TSAD tasks demonstrate that ANOMSEER consistently outperforms strong MLLM baselines (e.g., GPT-4o) in detection accuracy and localization precision, unifying detection, categorization, and reasoning. Critically, it produces fine-grained, plausible reasoning traces grounded in detailed time-series evidence, achieving faithful and verifiable interpretations in time-series anomaly detection.

## 2. Related Work

**Time series anomaly detection** is a critical task in domains like healthcare, aiming to detect deviations from normal temporal patterns (Wu et al., 2025; Shentu et al., 2024). Traditional methods rely on statistical techniques and machine learning methods (e.g., Z-score (Bhatnagar et al., 2021), Isolation Forest (Liu et al., 2008) and One-Class SVM (Schölkopf et al., 1999)), while recent advances use deep models such as Autoencoders (Zong et al., 2018; Park et al., 2018) for reconstruction- or prediction-based detection. Despite their effectiveness, these models struggle in industrial settings due to the scarcity of anomaly data, limiting generalization. To address this, recent efforts explore pre-trained (Zhou et al., 2023; Zhang et al., 2025a) and time-series foundation models (Goswami et al., 2024; Gao et al., 2024) for zero- and few-shot detection. However, these approaches are primarily optimized for accuracy, *lacking* the ability to analyze anomaly types, reason about temporal patterns, and explain why a given sample is anomalous.

**Time-series reasoning with LLMs** is an emerging research

frontier (Kong et al., 2026). To enable LLMs to perform time-series analysis, researchers have primarily explored two input strategies: prompting with numerical data (Al-negheimish et al., 2024) or visual representations (Zhuang et al., 2024; He et al., 2025; Xu et al., 2025; Zhou & Yu, 2024). While the visual approach, feeding plots into MLLMs such as GPT-4o, is often more token-efficient, its effectiveness is limited by the fact that these models are not explicitly trained on time-series visualizations. To instill temporal understanding, recent works have primarily relied on integrating classical modules (Chen et al., 2025; Liu et al., 2025a), employing auxiliary techniques (He et al., 2025; Zhuang et al., 2024), or large-scale SFT (Yang et al., 2025). An alternative and promising path involves RL to promote structured problem-solving, as seen in DeepSeek-R1 (Guo et al., 2025). Building on this, recent work such as TimeMaster (Zhang et al., 2025b) trains MLLMs for classification tasks by combining SFT with RL to enable interpretable temporal reasoning over visualized series. Nevertheless, RL for enhancing anomaly detection in MLLMs remains underexplored. In this paper, we show that vanilla RL struggles to detect subtle anomalies and propose a new method to mitigate this limitation.

## 3. Preliminary

**Time-series anomaly detection.** Time-series anomaly detection (TSAD) aims to identify abnormal patterns within temporal data. Following standard practice (Zhou & Yu, 2024), we use $\mathbf{X} = \{\mathbf{x}_t\}_{t=1}^{T}$ to denote a univariate time series of length $T$, where each observation $\mathbf{x}_t \in \mathbb{R}$ is sampled at regular intervals and may correspond to either normal or anomalous behavior. Anomalies are defined as continuous intervals of data points that deviate significantly from the expected pattern. They can be categorized into point-wise anomalies (contextual point and global point) and range-wise anomalies (trend, shapelet, and seasonal), resulting in five types in total. Formally, let $\mathcal{A} = \{(t_s^{(i)}, t_e^{(i)})\}_{i=1}^{k}$ denote the set of anomalous intervals, where $1 \leq t_s^{(i)} \leq t_e^{(i)} \leq T$. Each tuple $(t_s^{(i)}, t_e^{(i)})$ specifies the start and end indices of the $i$-th anomalous segment; in particular, $t_s^{(i)} = t_e^{(i)}$ denotes a single-point anomaly. The primary goal of TSAD is to infer the set $\mathcal{A}$ with high accuracy.

**Multimodal time-series formulation.** To enable MLLMs to perform time-series anomaly detection, the input of the MLLM consists of the time-series input $\mathbf{X}$ and context prompt $\mathbf{c}$ that encodes domain knowledge, natural-language instructions, or task-specific queries to guide the model's reasoning process. To enable multimodal processing, we follow the *visualization input strategy* (Liu et al., 2024; Xu et al., 2025; Zhang et al., 2025b), rendering the raw time series into a line-plot image $\mathbf{X} \rightarrow \mathbf{I}$ and then feeding it to the MLLM's vision encoder. This approach allows the

model to leverage its pre-trained visual reasoning abilities on a representation that is both compact and semantically rich (Xu et al., 2025; Xie et al., 2024).

**Multimodal LLM inference.** We define a time-series MLLM $\pi_\theta$ (parameterized by $\theta$) that specifies a conditional distribution over an output sequence $\mathbf{y} = \{y_1, y_2, \ldots, y_N\}$, where each token $y_n$ may correspond to an anomaly label, an interval boundary, or a natural-language reasoning. Given the rendered time-series data $\mathbf{I}$ and textual context $\mathbf{c}$, the model generates outputs autoregressively: $\pi_\theta(\mathbf{y} \mid \mathbf{I}, \mathbf{c}) = \prod_{n=1}^{N} \pi_\theta(y_n \mid y_{<n}, \mathbf{I}, \mathbf{c})$. This formulation unifies reasoning, explanation and detection in a single generative process, allowing the model to produce structured outputs that are both context-aware and interpretable.

## 4. Methodology

Time-series MLLMs often rely on coarse visual heuristics and fail to produce numerically grounded, fine-grained reasoning for TSAD. This weakness limits their ability to detect subtle anomalies such as frequency shifts or small trend drifts in complex time-series data. To address this, we introduce ANOMSEER, a novel MLLM post-training approach for TSAD that couples classical time-series statistical rigor with the expressive reasoning ability of MLLMs. ANOM-SEER is trained with two key components: (1) *expert chain-of-thought (ExpCoT)*, which generates structured, expert-like reasoning traces from ground-truth time series using statistical diagnostics (e.g., histogram-based outlier scores, FFT, matrix profile); and (2) *time-series grounded policy optimization (TimerPO)*, a new RL algorithm that leverages ExpCoT to establish the corrective, orthogonal advantages to refine reasoning without overriding the detection objective. Figure 2 presents an overview of ANOMSEER. In the remainder of this section, we will detail the design of Exp-CoT (Section 4.1) and the TimerPO optimization algorithm (Section 4.2), and discuss how they jointly enable accurate, interpretable, and numerically faithful anomaly detection.

### 4.1. Expert Chain-of-Thought Generation

To ground the reasoning of time-series MLLM with classical time-series detailed analysis for TSAD, we introduce the *expert chain-of-thought (ExpCoT)* trace, a structured reasoning that mirrors the stepwise detection of a human analyst. Unlike an LLM-generated CoT, which may rely on heuristic pattern matching, ExpCoT is grounded in systematically derived, quantitatively verifiable evidence. ExpCoT is generated per instance, starting from ground-truth annotations. We apply classical statistical and signal-processing techniques to extract descriptive statistics, candidate anomaly categories, and precise temporal localization. This trace delivers rich, multi-dimensional guidance that goes beyond a simple correct/incorrect signal, encouraging fine-grained

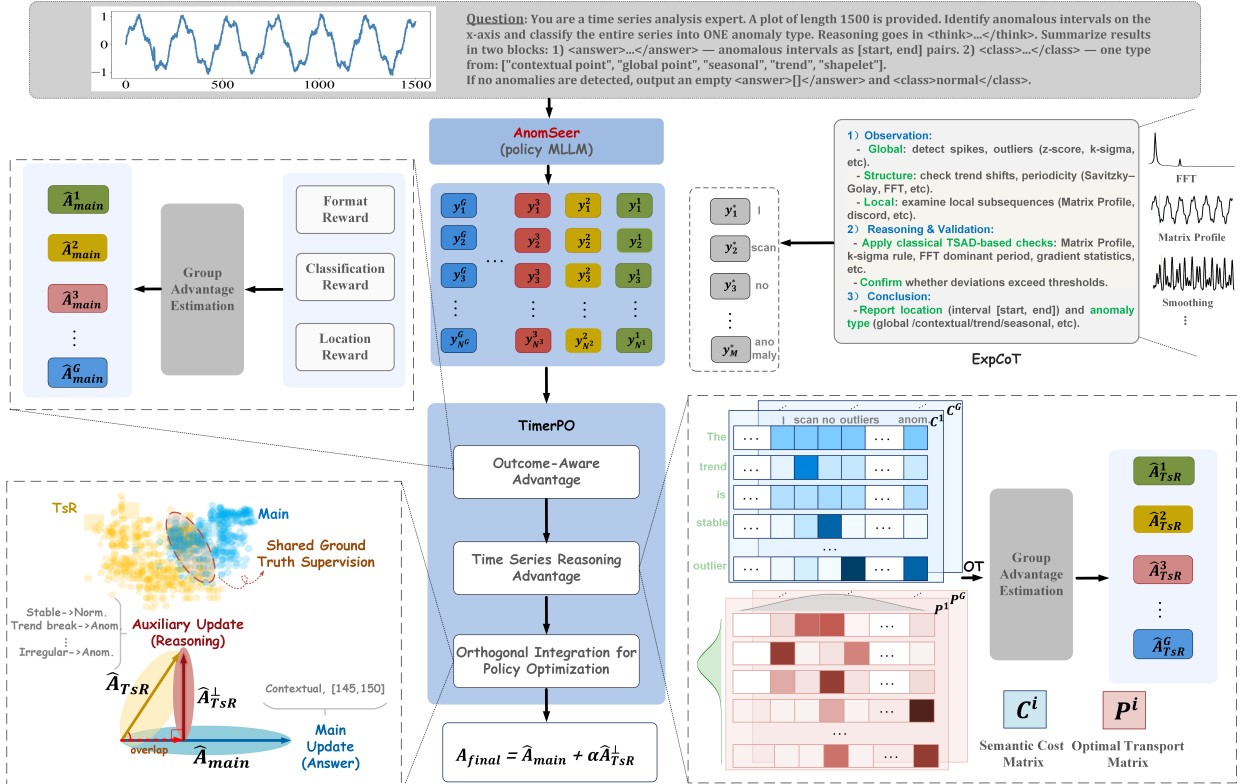

*Figure 2.* The overall framework of ANOMSEER. ANOMSEER first generates ExpCoT reasoning traces $\mathbf{y}^*$ from the time-series data based on classical TSAD techniques (e.g., FFT). TimerPO then computes the outcome-aware advantage and leverages optimal transport to compute the time-series reasoning advantage, which is orthogonally integrated into policy optimization to ensure stable training and improved reasoning quality.

and interpretable reasoning.

Crucially, ExpCoT adheres to a disciplined *three-stage* reasoning path (*Observation → Reasoning & Validation → Conclusion*), closely mirroring the stepwise process of human analytical reasoning.

> **Observation**: systematically progress from global patterns to local irregularities across multiple views, surfacing candidate anomalies prior to formal testing.

The *"Observation"* stage performs a hierarchical scan of the time series $\mathbf{X}$ to extract preliminary statistical features. (**1**) *Global Scan:* We first assess extreme values by examining the global data distribution via a histogram-based outlier score (Goldstein & Dengel, 2012). (**2**) *Structural Scan:* If no global outliers are present, we analyze fundamental properties such as trend stability using smoothed gradients (Thill et al., 2017) and periodicity via FFT-based frequency analysis (Ren et al., 2019). (**3**) *Local Scan:* If the series appears structurally stable, we perform a localized search for dissimilar subsequences (discords) using the Matrix Profile (Yeh et al., 2016). This fine-grained scan provides the key statistical features that guide the subsequent detection process.

> **Reasoning & Validation**: use classical TSAD techniques to formalize ground truth as fine-grained, testable claims, supported by targeted quantitative time-series analysis.

The *"Reasoning & Validation"* stage establishes a causal link between preliminary observations and formal statistical evidence of anomalies. First, it leverages the ground-truth anomaly type to align statistical markers with visual patterns (e.g., "A sharp spike around $t \approx 150$ deviates significantly from the rest of the data, suggesting a contextual anomaly"). This classification then guides the selection of a targeted statistical method for validation; for example, a suspected trend shift is validated using gradient analysis (Thill et al., 2017), while the aforementioned contextual anomaly is confirmed by its Matrix Profile score (Yeh et al., 2016). The numerical outcome is translated into a natural language explanation (e.g., "The discord's z-score of 4.2 at timestamp 145 exceeds the 3-sigma threshold, confirming a significant pattern deviation").

> **Conclusion**: integrate multi-dimensional insights and fine-grained evidence into a precise, defensible anomaly judgment.

The final *"Conclusion"* stage synthesizes the findings into a conclusive summary. It integrates the multi-dimensional understanding from the *"Observation"* stage with the detailed, quantitative evidence from the *"Reasoning & Validation"* stage to deliver a definitive judgment, e.g., "Therefore, the detected anomaly is a contextual point, located in the interval [145, 150]".

In summary, as shown in Figure 2, ExpCoT provides a structured reasoning trace that embeds analytical rigor and numerically grounded logic. This makes it particularly effective for identifying subtle anomalies and offers fine-grained, informed guidance for subsequent MLLM training. See examples of ExpCoT in Appendix B.2.

## 4.2. Time-Series Grounded Policy Optimization

To leverage ExpCoT and enable the reasoning of MLLM grounded in fine-grained time-series analysis, we introduce TimerPO, a novel RL method building upon Group Relative Policy Optimization (GRPO) (Shao et al., 2024). We begin with the vanilla GRPO formulation. Given the rendered time-series instance $\mathbf{I}$ and textual context $\mathbf{c}$, the model produces a *group* of candidate responses $\mathcal{G} = \{\mathbf{y}^1, \mathbf{y}^2, ..., \mathbf{y}^G\}$ where $G$ denotes the group size. This group-based generation enables pairwise relative reward comparisons, which are subsequently used to compute group-aware advantages.

**Outcome-Aware Advantage.** For each generated response $\mathbf{y}^i \in \mathcal{G}$, the task reward is a weighted sum of (i) a format reward $r^{\mathrm{fmt},i} \in \{0, 1\}$ that checks if the predefined output format of time-series MLLM is valid, (ii) a classification reward $r^{\mathrm{cls},i}$ for anomaly type accuracy and (iii) a detection location reward $r^{\mathrm{loc},i}$, which integrates common anomaly-detection metrics (Zhou & Yu, 2024):

$$r^i = \lambda^{\mathrm{fmt}} r^{\mathrm{fmt},i} + \lambda^{\mathrm{cls}} r^{\mathrm{cls},i} + \lambda^{\mathrm{loc}} r^{\mathrm{loc},i}, \quad (1)$$

where $\lambda^{\mathrm{fmt}}, \lambda^{\mathrm{cls}}, \lambda^{\mathrm{loc}}$ are tunable weights. To stabilize optimization, rewards are normalized within each group, yielding the main advantage:

$$\widehat{A}^i_{\mathrm{main}} = \frac{r^i - \mu_r}{\sigma_r + \varepsilon}, \quad \mu_r = \frac{1}{G} \sum_{i=1}^G r^i, \quad (2)$$

where $\sigma_r^2 = \frac{1}{G} \sum_{i=1}^G (r^i - \mu_r)^2$. The vectorized form $\widehat{A}_{\mathrm{main}} = (\widehat{A}^1_{\mathrm{main}}, \ldots, \widehat{A}^G_{\mathrm{main}})^\top \in \mathbb{R}^G$ serves as the normalized baseline signal for subsequent policy updates. However, such outcome-aware advantages risk encouraging coarse, heuristic reasoning for time series data (e.g., detecting only obvious outliers while ignoring subtle but meaningful temporal patterns).

**Time-Series Reasoning Advantage.** To explicitly encourage fine-grained reasoning, TimerPO leverages the Optimal Transport (OT) (Villani et al., 2008; Li et al., 2024) to quantify the semantic alignment between a model's reasoning trace $\mathbf{y}^i = \{y^i_1, \ldots, y^i_{N^i}\}$ and the corresponding ExpCoT's reasoning trace $\mathbf{y}^\star = \{y^\star_1, \ldots, y^\star_M\}$ where $N^i$ and $M$ are their lengths. Given $\mathbf{y}^i$ and $\mathbf{y}^\star$, we extract the final-layer embeddings from the MLLM $\pi_\theta$, obtaining embedding vectors $\mathbf{e}^i$ for $\mathbf{y}^i$ and $\mathbf{e}^\star$ for $\mathbf{y}^\star$. We then construct a semantic cost matrix $\mathbf{C}^i \in \mathbb{R}^{N^i \times M}$ whose $(n, m)$-th entry measures the cosine distance between token embeddings:

$$C^i_{nm} = 1 - \frac{\mathbf{e}^i_n \cdot \mathbf{e}^\star_m}{\|\mathbf{e}^i_n\| \|\mathbf{e}^\star_m\|}, \ n = 1, \ldots, N^i, \ m = 1, \ldots, M. \quad (3)$$

Let $\mathbf{u}^i \in \Delta^{N^i-1}$ and $\mathbf{v} \in \Delta^{M-1}$ denote the marginal distributions over token positions for the model and the corresponding ExpCoT trace, obtained by normalizing their generation probabilities. The OT distance for response $\mathbf{y}^i$ is defined by

$$
\begin{aligned}
W^i &= \min_{\mathbf{P}^i \in \Pi(\mathbf{u}^i, \mathbf{v})} \langle \mathbf{P}^i, \mathbf{C}^i \rangle_F, \\
\Pi(\mathbf{u}^i, \mathbf{v}) &= \{\mathbf{P}^i \geq 0 \mid \mathbf{P}^i \mathbb{1}_M = \mathbf{u}^i, (\mathbf{P}^i)^\top \mathbb{1}_{N^i} = \mathbf{v}\},
\end{aligned} \quad (4)
$$

where $\langle \cdot, \cdot \rangle_F$ is the Frobenius product, and $W^i$ measures the minimal semantic effort required to transform the model's reasoning distribution into the ExpCoT distribution. In practice, we approximate the solution of Equation (4) with the entropic-regularized Sinkhorn–Knopp (Cuturi, 2013) for efficiency and smoothness. Then, we use $r^i_{\mathrm{TsR}} = \exp(-W^i/\tau)$ as the reasoning reward and obtain the *time-series reasoning advantage*:

$$\widehat{A}^i_{\mathrm{TsR}} = \frac{r^i_{\mathrm{TsR}} - \mu_{\mathrm{TsR}}}{\sigma_{\mathrm{TsR}} + \varepsilon}, \quad \mu_{\mathrm{TsR}} = \frac{1}{G} \sum_{i=1}^G r^i_{\mathrm{TsR}}, \quad (5)$$

where $\sigma^2_{\mathrm{TsR}} = \frac{1}{G} \sum_{i=1}^G (r^i_{\mathrm{TsR}} - \mu_{\mathrm{TsR}})^2$. By collecting the values across the group $\mathcal{G}$, we obtain $\widehat{A}_{\mathrm{TsR}} = (\widehat{A}^1_{\mathrm{TsR}}, \ldots, \widehat{A}^G_{\mathrm{TsR}})^\top \in \mathbb{R}^G$, which serves as a relative measure of reasoning quality.

**Orthogonal Integration for Policy Optimization.** A naive combination of task and reasoning rewards risks interference, as ExpCoT guidance may overlap with the primary detection objective under shared ground truth supervision. To avoid this, TimerPO orthogonalizes the time-series grounded advantage with respect to the main advantage, retaining only the complementary part:

$$\widehat{A}^\perp_{\mathrm{TsR}} = \widehat{A}_{\mathrm{TsR}} - \frac{\langle \widehat{A}_{\mathrm{TsR}}, \widehat{A}_{\mathrm{main}} \rangle}{\|\widehat{A}_{\mathrm{main}}\|_2^2 + \varepsilon} \widehat{A}_{\mathrm{main}}. \quad (6)$$

We then compose the final advantage for each response by

$$A^i_{\mathrm{final}} = \widehat{A}^i_{\mathrm{main}} + \alpha \left(\widehat{A}^\perp_{\mathrm{TsR}}\right)^i, \qquad i = 1, \ldots, G, \quad (7)$$

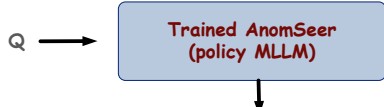

**Q** →

Answer: <think> The series presents a repeating *waveform* with consistent cycles. Most *segments* preserve a similar rise-and-fall shape with stable *amplitude*. **Around timestamp 149, the curve departs from this rhythm: a short *segment* shows an abrupt spike and reduced stability compared with adjacent cycles.** The *values* remain within the normal range, yet the *interval 140–155* breaks the surrounding pattern. This irregularity constitutes a contextual anomaly driven by disruption of the local sequence. </think>
<class>contextual point</class>
<answer>[[140,155]]</answer>

*Figure 3.* An example of TSAD reasoning by ANOMSEER at inference. The model runs fully end-to-end, without relying on ExpCoT traces or classical detectors.

where $\alpha$ is a hyperparameter controlling the strength of the reasoning refinement. This composite advantage, $A_{\text{final}}^i$, then drives the policy update by replacing the standard normalized advantage in the clipped objective function:

$$\mathcal{L}(\theta) = \frac{1}{G} \sum_{i=1}^{G} \frac{1}{|\mathbf{y}^i|} \sum_{n=1}^{|\mathbf{y}^i|} \min\left(\rho_n^i A_{\text{final}}^i, \ \tilde{A}_n^i\right) - \beta \, \text{KL}\big[\pi_\theta \| \pi_{\text{ref}}\big],$$

(8)

where $\rho_n^i$ is the importance ratio for the $n$-th token of response $\mathbf{y}^i$, and $\tilde{A}_n^i = \text{clip}(\rho_n^i, 1-\epsilon, 1+\epsilon)\, A_{\text{final}}^i$, with $\epsilon$ and $\beta$ denoting the PPO clipping and KL coefficients, respectively. By operating at the advantage level, TimerPO offers a stable mechanism to instill ExpCoT reasoning, enhancing the model's analytical precision while keeping the primary detection update direction unchanged.

**Overall.** ANOMSEER employs the pure RL training strategy to enhance MLLMs without SFT as a cold-start or any modifications to the model architecture. During training, we first construct ExpCoT using the analytical rigor of traditional TSAD methods, and subsequently refine the model's policy using orthogonalized time-series reasoning advantages through TimerPO. This simple yet effective integrated design efficiently instills expert knowledge into the pre-trained model within a single reinforcement learning phase. At inference time, ANOMSEER operates in a fully end-to-end manner, requiring no external components or incurring any additional token overhead. As shown in Figure 3, the trained ANOMSEER receives the question and produces outputs that include step-by-step analysis, anomaly type classification, and precise interval localization. Appendix A provides the pseudocode for the overall ANOMSEER procedure.

## 5. Experiments

**Benchmarks.** To evaluate the performance and generalization ability of ANOMSEER, we consider three diverse TSAD benchmarks: (**1**) *AnomLLM* (Zhou & Yu, 2024), a synthetic dataset containing frequency, trend, out-of-range and point anomalies[2]; (**2**) *VisualTimeAnomaly* (Xu et al.,

---

[2]In AnomLLM, contextual *frequency*, *trend*, and *point* anomalies are harder as they require contextual awareness, and *range*

2025), a mixed synthetic–real, image-based benchmark covering a broader spectrum of anomaly types[3]; and (**3**) *TSB-UAD* (Paparrizos et al., 2022; Qiu et al., 2025), a real-world univariate collection from domains such as ECG and web traffic, with diverse anomaly types, ratios, and sequence lengths. Training is conducted solely on the synthetic Anom-LLM benchmark (3,200 instances), ensuring clean, high-fidelity ExpCoT supervision. Evaluation is then performed on the test sets of AnomLLM, the mixed real-world Visual-TimeAnomaly, and TSB-UAD, providing a rigorous test of generalization to diverse, previously unseen anomalies.

**Baselines.** We compare against both commercial (GPT-4o, GPT-4o-mini, Gemini-2.5-Pro, Gemini-2.5-Flash) and open-source MLLMs (Qwen2.5-VL-72B/32B/7B/3B-Instruct), as well as two representative LLM-based temporal reasoning baselines: *SigLLM* (GPT-3.5-based) (Alnegheimish et al., 2024) and *TimeMaster* (Qwen2.5-VL-3B-based, trained with SFT and GRPO) (Zhang et al., 2025b). We further compare against SFT baselines: Qwen2.5-VL-3B-SFT3.2k, fine-tuned on 3,200 instances, and Qwen2.5-VL-3B-SFT32k, fine-tuned on 32,000 instances.

**Metrics.** We report both anomaly-type classification accuracy and label-based metrics for localization performance, including Affinity-Precision (P), Affinity-Recall (R), and Affinity-F1 (F1), following the definitions in Huet et al. (2022). These metrics are suitable because LLMs generate discrete anomalous intervals, which can be converted into binary labels rather than continuous scores, and they better capture the temporal consistency of anomaly detection (Zhou & Yu, 2024; Xu et al., 2025).

**Hyperparameters.** We build ANOMSEER on Qwen2.5-VL-3B/7B-Instruct (Bai et al., 2025). Following Zhang et al. (2025b), we set the group size $G = 5$ and the PPO clipping $\epsilon = 0.2$. The reward weights are empirically chosen as $\lambda^{\text{fmt}} = 0.1$, $\lambda^{\text{cls}} = 0.2$, and $\lambda^{\text{loc}} = 0.7$. TimerPO's reasoning advantage weight is fixed at $\alpha = 0.3$. More experimental details are provided in Appendix C.

### 5.1. Main Results

As shown in Table 1, ANOMSEER consistently achieves state-of-the-art results across all anomaly detection tasks on the AnomLLM benchmark. Remarkably, even at a lightweight 3B scale, our model substantially outperforms much larger and more resource-intensive MLLMs such as GPT-4o and Gemini-2.5-Pro in both anomaly type classification and Affinity-F1 metrics, and its performance further improves with the 7B variant. We also observe that simply

---

anomalies are easier as they show obvious global point deviations.

[3]In VisualTimeAnomaly, *range-wise* anomalies (shapelet, seasonal, and trend) are generally easier, while *point-wise contextual* and *global* anomalies, which manifest as subtle and dispersed single points, are harder.

*Table 1.* Performance comparison on the AnomLLM test dataset. Results are reported as the mean and standard deviation over three runs for anomaly classification accuracy (%) and location detection accuracy metrics (%): Affinity-Precision (P), Affinity-Recall (R), and Affinity-F1 (F1).

| Modality | Type | Method | Classification Accuracy | Location Frequency P | R | F1 | Trend P | R | F1 | Range P | R | F1 | Point P | R | F1 | Avg F1 |
|---|---|---|---|---|---|---|---|---|---|---|---|---|---|---|---|---|
| *Commercial* | | | | | | | | | | | | | | | | |
| Image+Text | Prompting | GPT-4o | $17.2_{\pm1.1}$ | $11.1_{\pm0.2}$ | $10.8_{\pm0.2}$ | $10.9_{\pm0.2}$ | $40.3_{\pm0.4}$ | $48.4_{\pm0.1}$ | $43.5_{\pm0.2}$ | $55.0_{\pm0.5}$ | $61.8_{\pm0.5}$ | $57.0_{\pm0.5}$ | $51.5_{\pm0.2}$ | $58.8_{\pm0.2}$ | $53.4_{\pm0.2}$ | 41.2 |
| Image+Text | Prompting | GPT-4o-mini | $17.8_{\pm1.2}$ | $10.3_{\pm0.1}$ | $10.1_{\pm0.1}$ | $10.2_{\pm0.1}$ | $19.4_{\pm0.2}$ | $29.4_{\pm0.2}$ | $23.2_{\pm0.2}$ | $48.0_{\pm0.1}$ | $58.6_{\pm0.1}$ | $51.3_{\pm0.1}$ | $51.3_{\pm0.1}$ | $58.2_{\pm0.1}$ | $52.7_{\pm0.1}$ | 34.4 |
| Image+Text | Prompting | Gemini-2.5-Flash | $10.0_{\pm0.5}$ | $21.4_{\pm0.1}$ | $16.6_{\pm0.1}$ | $17.9_{\pm0.1}$ | $34.6_{\pm0.1}$ | $36.0_{\pm0.1}$ | $35.2_{\pm0.1}$ | $76.0_{\pm0.1}$ | $78.9_{\pm0.1}$ | $76.7_{\pm0.1}$ | $76.4_{\pm0.5}$ | $74.9_{\pm0.5}$ | $74.9_{\pm0.5}$ | 51.2 |
| Image+Text | Prompting | Gemini-2.5-Pro | $12.6_{\pm0.1}$ | $17.4_{\pm0.1}$ | $22.0_{\pm0.1}$ | $19.1_{\pm0.3}$ | $58.8_{\pm0.1}$ | $60.0_{\pm0.1}$ | $59.0_{\pm0.2}$ | $79.4_{\pm0.1}$ | $83.2_{\pm0.5}$ | $81.3_{\pm0.4}$ | $76.1_{\pm0.5}$ | $74.4_{\pm0.4}$ | $74.5_{\pm0.5}$ | 58.5 |
| Numerical+Text | Prompting | SigLLM (GPT-3.5) | \ | $16.9_{\pm0.5}$ | $14.9_{\pm0.5}$ | $15.8_{\pm0.5}$ | $20.3_{\pm0.5}$ | $20.5_{\pm0.5}$ | $19.6_{\pm0.1}$ | $67.8_{\pm0.1}$ | $67.5_{\pm0.1}$ | $67.7_{\pm0.1}$ | $34.4_{\pm0.1}$ | $38.6_{\pm0.1}$ | $36.2_{\pm0.1}$ | 34.8 |
| *Open-source* | | | | | | | | | | | | | | | | |
| Image+Text | Prompting | Qwen2.5-VL-72B-Instruct | $14.6_{\pm0.5}$ | $40.2_{\pm0.1}$ | $28.3_{\pm0.1}$ | $31.4_{\pm0.1}$ | $30.8_{\pm0.2}$ | $33.8_{\pm0.1}$ | $32.1_{\pm0.7}$ | $76.8_{\pm0.1}$ | $73.9_{\pm0.1}$ | $74.6_{\pm0.1}$ | $63.2_{\pm0.5}$ | $64.6_{\pm0.3}$ | $62.7_{\pm0.1}$ | 50.2 |
| Image+Text | Prompting | Qwen2.5-VL-32B-Instruct | $10.2_{\pm0.5}$ | $19.3_{\pm0.4}$ | $20.2_{\pm0.5}$ | $18.9_{\pm0.2}$ | $34.3_{\pm0.5}$ | $37.2_{\pm0.2}$ | $35.5_{\pm0.5}$ | $72.0_{\pm0.5}$ | $70.3_{\pm0.5}$ | $70.7_{\pm0.5}$ | $55.9_{\pm0.5}$ | $59.3_{\pm0.5}$ | $56.7_{\pm0.5}$ | 45.5 |
| Image+Text | Prompting | Qwen2.5-VL-7B-Instruct | $25.3_{\pm0.2}$ | $18.5_{\pm0.4}$ | $16.4_{\pm0.1}$ | $16.8_{\pm0.2}$ | $52.7_{\pm0.1}$ | $53.8_{\pm0.2}$ | $53.1_{\pm0.3}$ | $48.6_{\pm0.4}$ | $45.4_{\pm0.1}$ | $46.4_{\pm0.1}$ | $61.9_{\pm0.1}$ | $58.6_{\pm0.1}$ | $59.0_{\pm0.1}$ | 43.8 |
| Image+Text | Prompting | Qwen2.5-VL-3B-Instruct | $11.4_{\pm0.1}$ | $7.1_{\pm0.1}$ | $9.3_{\pm0.2}$ | $7.9_{\pm0.4}$ | $17.4_{\pm0.1}$ | $22.0_{\pm0.5}$ | $19.1_{\pm0.1}$ | $29.7_{\pm0.1}$ | $31.3_{\pm0.1}$ | $29.7_{\pm0.1}$ | $34.3_{\pm0.5}$ | $40.6_{\pm0.5}$ | $36.2_{\pm0.1}$ | 23.2 |
| Image+Text | Training | Qwen2.5-VL-3B-SFT3.2K | $29.7_{\pm0.1}$ | $19.0_{\pm0.1}$ | $24.4_{\pm0.5}$ | $19.7_{\pm0.1}$ | $29.7_{\pm0.2}$ | $34.8_{\pm0.1}$ | $30.2_{\pm0.2}$ | $40.1_{\pm0.1}$ | $48.1_{\pm0.1}$ | $40.8_{\pm0.1}$ | $40.4_{\pm0.5}$ | $49.3_{\pm0.1}$ | $40.6_{\pm0.2}$ | 32.8 |
| Image+Text | Training | Qwen2.5-VL-3B-SFT32K | $35.6_{\pm0.5}$ | $12.0_{\pm0.5}$ | $14.3_{\pm0.1}$ | $12.4_{\pm0.1}$ | $57.6_{\pm0.1}$ | $57.5_{\pm0.2}$ | $57.4_{\pm0.1}$ | $40.4_{\pm0.1}$ | $52.1_{\pm0.1}$ | $41.3_{\pm0.1}$ | $44.3_{\pm0.5}$ | $58.3_{\pm0.2}$ | $46.3_{\pm0.5}$ | 39.4 |
| Image+Text | Training | TimeMaster-3B | $57.9_{\pm0.6}$ | $57.3_{\pm0.5}$ | $50.3_{\pm0.1}$ | $51.4_{\pm0.2}$ | $76.0_{\pm0.5}$ | $77.3_{\pm0.1}$ | $76.6_{\pm0.5}$ | $77.8_{\pm0.5}$ | $83.5_{\pm0.1}$ | $80.1_{\pm0.1}$ | $77.7_{\pm0.5}$ | $82.1_{\pm0.1}$ | $79.6_{\pm0.5}$ | 71.9 |
| Image+Text | Training | ANOMSEER-3B (Ours) | $62.8_{\pm0.5}$ | $63.7_{\pm0.5}$ | $58.4_{\pm0.5}$ | $58.9_{\pm0.5}$ | $84.2_{\pm0.2}$ | $85.9_{\pm0.1}$ | $84.9_{\pm0.1}$ | $83.3_{\pm0.3}$ | $89.2_{\pm0.1}$ | $85.6_{\pm0.1}$ | $86.0_{\pm0.1}$ | $90.3_{\pm0.1}$ | $87.8_{\pm0.1}$ | 79.3 |
| Image+Text | Training | ANOMSEER-7B (Ours) | $65.0_{\pm0.5}$ | $68.3_{\pm0.5}$ | $59.4_{\pm0.2}$ | $60.8_{\pm0.2}$ | $86.6_{\pm0.5}$ | $89.0_{\pm0.5}$ | $87.7_{\pm0.5}$ | $91.6_{\pm0.1}$ | $97.8_{\pm0.4}$ | $94.3_{\pm0.1}$ | $93.4_{\pm0.1}$ | $96.9_{\pm0.5}$ | $94.9_{\pm0.1}$ | 84.4 |

*Table 2.* Ablation study on different components of ANOMSEER-3B using Affinity F1 score (%).

| Components | | | Anomaly Scenarios | | | |
|---|---|---|---|---|---|---|
| ExpCoT | $\widehat{A}^{\perp}_{\text{TsR}}$ | Orth | Frequency | Trend | Range | Point |
| ✗ | ✓ | ✓ | 49.8 | 79.5 | 84.4 | 86.1 |
| ✓ | ✓ | ✗ | 53.5 | 81.1 | 83.5 | 85.4 |
| ✗ | ✗ | ✗ | 50.4 | 77.8 | 81.8 | 80.6 |
| ✓ | ✓ | ✓ | **58.9** | **84.9** | **85.6** | **87.8** |

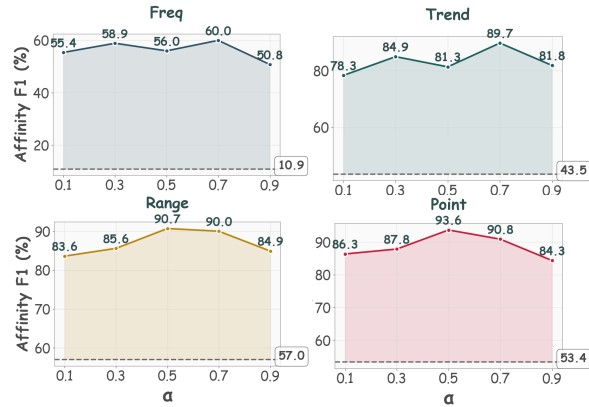

*Figure 4.* Hyperparameter sensitivity analysis on $\alpha$, comparing our method with the GPT-4o baseline (grey dashed line).

increasing the amount of SFT data yields only marginal gains, even with $10\times$ more SFT data (32k instances), performance still falls short of ANOMSEER. One possible reason is that SFT emphasizes only positive reasoning paths while neglecting negative ones, leading the model to develop only a shallow understanding rather than genuinely learning. Notably, for numerically subtle anomalies such as frequency shifts, ANOMSEER maintains a clear advantage, whereas GRPO-trained MLLMs like TimeMaster continue to lag behind. This result suggests that globally verifiable RL objectives alone are insufficient for modeling fine-grained temporal variations, whereas our ANOMSEER explicitly encourages fine-grained temporal reasoning that leads to more accurate anomaly detection.

## 5.2. Ablation Study and Hyperparameter Analysis

We next conduct a detailed ablation study together with a hyperparameter sensitivity analysis. Table 2 provides several key takeaways. First, we replace ExpCoT with CoT generated by GPT-4o, which leads to a marked degradation, particularly on challenging frequency anomalies. This demonstrates that generic CoT supervision imparts mere surface-level fluency rather than in-depth temporal reasoning. It further highlights a crucial insight: the analytical

rigor of classical methods is not obsolete, but rather a valuable resource for shaping the next generation of truly capable time-series MLLMs. Second, removing the orthogonalization mechanism causes a moderate drop in performance, underscoring its crucial role in mitigating spurious correlations between reasoning quality and task success. Third, eliminating all components reduces the method to a vanilla GRPO setup and yields the worst average performance, confirming that outcome-based rewards alone are insufficient to foster the fine-grained anomaly detection skills required for complex TSAD.

Figure 4 presents the effect of varying the temporal reasoning weight $\alpha$ in our TimerPO objective. Across all anomaly types, ANOMSEER maintains a substantial margin over the GPT-4o baseline (grey dashed line), showing that even under suboptimal $\alpha$ values, the integration of structured temporal reasoning signals offers clear benefits. The model remains

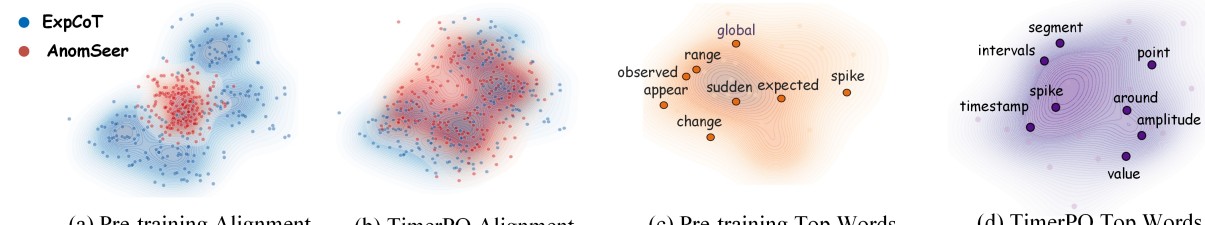

(a) Pre-training Alignment     (b) TimerPO Alignment     (c) Pre-training Top Words     (d) TimerPO Top Words

*Figure 5.* Comparison of distribution alignment between ExpCoT (blue) and ANOMSEER (red) outputs, as well as token usage before and after applying TimerPO.

relatively robust within the range $\alpha \in [0.3, 0.7]$, where performance is stable and near-optimal for frequency, trend, range, and point anomalies alike. This highlights the importance of balancing outcome-level and reasoning-level rewards: too small a weight diminishes the impact of explicit reasoning supervision, while too large a weight can overshadow task-level alignment, leading to slight degradation. In practice, $\alpha = 0.3$ works well as a default, though dataset-specific tuning may yield more gains.

### 5.3. Effect of TimerPO on Reasoning Pattern

To show that ANOMSEER enables time-series MLLMs reasoning grounded in fine-grained statistics, we analyze the effect of TimerPO on distributional alignment and linguistic usage before and after RL training, as shown in Figure 5. Panels (a)-(b) illustrate that, prior to TimerPO, ExpCoT (blue) and ANOMSEER outputs (red) occupy noticeably divergent regions in the representation space, with the latter exhibiting a relatively narrow distribution. This mismatch highlights that the model's reasoning is overly global and lacks diversity. A similar trend is observed in token usage. In the pre-training stage (c), top words are generic and coarse-grained (e.g., global, sudden, change), reflecting surface-level anomaly descriptions. After TimerPO (d), the vocabulary shifts toward finer-grained and temporally grounded tokens (e.g., timestamp, intervals, amplitude), which better capture structured reasoning over time. Therefore, these results demonstrate that TimerPO not only improves distributional alignment with expert reasoning but also enriches the semantic granularity of reasoning traces, moving from broad anomaly descriptors to precise temporal markers. We also compare GRPO and our TimerPO-trained models in Appendix D.6, which further confirms the effectiveness of our method in enhancing temporal reasoning.

### 5.4. Generalization Performance

At last, we evaluate the generalization ability of ANOMSEER. We test the model (trained on the synthetic Anom-LLM) on two distinct and more challenging benchmarks: VisualTimeAnomaly (a hybrid synthetic-real dataset with richer anomaly types) and TSB-UAD (a real-world univari-

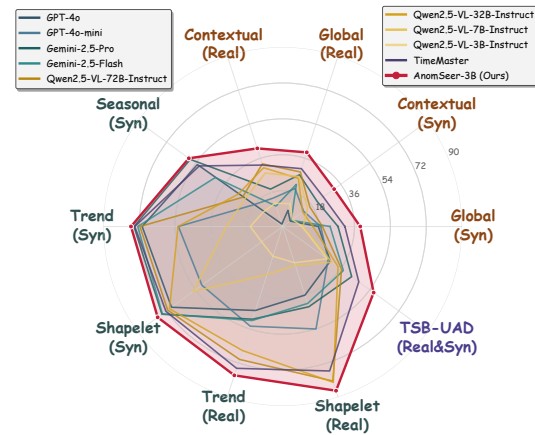

*Figure 6.* Comparison of model generalization performance (Affinity F1%) across point-wise tasks, range-wise tasks, and the real-world TSB-UAD benchmark.

ate collection). Importantly, shapelet anomalies represent a completely new category absent during training. Despite this, as shown in Figure 6, our method demonstrates strong accuracy on such cases. This ability to detect and explain shapelet anomalies shows that the model is not restricted to pattern memorization, but can generalize to qualitatively novel anomaly behaviors. Moreover, on point-wise contextual anomalies, a notably harder task that requires fine-grained discrimination, ANOMSEER delivers clear gains over baseline MLLMs, underscoring its ability to move beyond surface-level visual cues. Finally, on the TSB-UAD collection of real-world datasets, which spans diverse domains, ANOMSEER sustains its advantage and confirms that the improvements extend beyond synthetic benchmarks to practical anomaly detection scenarios. Overall, these results verify that our approach achieves not only high in-domain accuracy but also robust generalization to unseen and real-world anomalies.

## 6. Conclusions and Limitations

In this paper, we introduced ANOMSEER, an RL post-training method that enables multimodal LLMs to detect and reason about time-series anomalies in a fine-grained and accurate manner. By grounding MLLMs' reasoning

in the fine-grained, multi-dimensional evidence of classical TSAD, ANOMSEER attains state-of-the-art performance across diverse benchmarks. Beyond surpassing strong baselines such as GPT-4o in detection accuracy and localization, it delivers verifiable, detailed time-series explanations, elevating MLLMs from coarse visual heuristics to principled, testable analysis. Nevertheless, ANOMSEER was developed primarily on univariate time-series data in TSAD, and extending it to more complex multivariate scenarios remains an open direction. A potential solution is to reframe each variable as an image-like subrepresentation and then reason over its joint structure, enabling the model to capture both localized temporal patterns and cross-variable dependencies in a coherent manner. Another direction may be to explore how to incorporate external knowledge to better account for real-world events that drive anomaly dynamics.

## Acknowledgements

This work is supported in part by the National Key R&D Program of China (2024YFF0907701) and the Ministry of Education, Singapore, under its Academic Research Fund Tier 1 (RG101/24). Xu Guo thanks the support from Wallenberg-NTU Presidential Postdoctoral Fellowship.

We are grateful to Suyu Liu for sharing his expertise in optimal transport and multi-objective optimization, which significantly strengthened this work.

## Impact Statement

This paper presents work whose goal is to advance the field of Machine Learning. There are many potential societal consequences of our work, none which we feel must be specifically highlighted here.

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

# A. Pseudo Code

The training pipeline of ANOMSEER is provided as follows:

---

**Algorithm 1** Training Time-Series MLLMs with ANOMSEER

---

1: **Require:** Initial policy $\pi_{\theta_{\text{old}}}$, task distribution $p(\mathbf{X})$, discount factor $\gamma$, clipping parameter $\epsilon$, KL penalty $\beta$, group size $G$, ExpCoT generator, TimerPO hyperparameter $\alpha$
2: **for** each training iteration **do**
3:     Update old policy: $\theta_{\text{old}} \leftarrow \theta$
4:     // Data preparation phase
5:     Sample time-series $\mathbf{X} \sim p(\mathbf{X})$ and render visualization $I$
6:     Generate expert chain-of-thought $\mathbf{y}^\star \leftarrow \text{ExpCoT}(\mathbf{X})$
7:     Construct input $(\mathbf{I}, \mathbf{c})$
8:     // Advantage computation
9:     Sample group of responses $\mathcal{G} = \{\mathbf{y}^i \sim \pi_{\theta_{\text{old}}}(\cdot|\mathbf{I}, \mathbf{c})\}_{i=1}^G$
10:     **for** each $\mathbf{y}^i \in \mathcal{G}$ **do**
11:       Compute outcome reward: $r^i = \lambda^{\text{fmt}} r^{\text{fmt}, i} + \lambda^{\text{cls}} r^{\text{cls}, i} + \lambda^{\text{loc}} r^{\text{loc}, i}$
12:       Normalize to obtain outcome-aware advantage $\widehat{A}_{\text{main}}^i$ via Eq. (2)
13:       Compute semantic OT distance $W^i$ between $\mathbf{y}^i$ and $\mathbf{y}^\star$ via Eq. (4)
14:       Derive reasoning reward $r_i^{\text{TsR}} = \exp(-W^i/\tau)$ and normalize to $\widehat{A}_{\text{TsR}}^i$ via Eq. (5)
15:     **end for**
16:     // Orthogonal integration of advantages
17:     Compute orthogonalized reasoning advantage:

$$\widehat{A}_{\text{TsR}}^\perp = \widehat{A}_{\text{TsR}} - \frac{\langle \widehat{A}_{\text{TsR}}, \widehat{A}_{\text{main}} \rangle}{\|\widehat{A}_{\text{main}}\|_2^2 + \varepsilon} \widehat{A}_{\text{main}}$$

18:     Final advantage: $A_{\text{final}}^i = \widehat{A}_{\text{main}}^i + \alpha \left(\widehat{A}_{\text{TsR}}^\perp\right)^i$
19:     // Policy update
20:     Update $\theta$ by maximizing the TimerPO objective:

$$\mathcal{L}(\theta) = \frac{1}{G} \sum_{i=1}^G \frac{1}{|\mathbf{y}^i|} \sum_{n=1}^{|\mathbf{y}^i|} \min\left(\rho_n^i A_{\text{final}}^i, \ \text{clip}(\rho_n^i, 1 - \epsilon, 1 + \epsilon) A_{\text{final}}^i\right) - \beta \, \text{KL}\big[\pi_\theta \, \| \, \pi_{\text{ref}}\big],$$

21: **end for**

---

---

**Algorithm 2** Inference with ANOMSEER

---

1: **Require:** Trained policy $\pi_\theta$, input time series $\mathbf{X}$, instruction prompt $\mathbf{c}$
2: Render visualization: $\mathbf{I} \leftarrow \mathcal{R}(\mathbf{X})$
3: Construct model input: $(\mathbf{I}, \mathbf{c})$
4: // Forward inference
5: Generate model response: $\mathbf{y} \sim \pi_\theta(\cdot \mid \mathbf{I}, \mathbf{c})$
6: Get output $\mathbf{y}$ including anomaly type, location, and reasoning
7: **return** anomaly prediction results

---

# B. More Details of ANOMSEER

## B.1. Structured Output for Reasoning.

A key objective of ANOMSEER is to elicit *textual reasoning* that illuminates the model's analysis process. To achieve this, we enforce a structured output format to decouple the reasoning steps from the final prediction. The model is prompted to first articulate its analytical process within <think> </think> tags, provide the predicted anomaly category (e.g., trend, global, contextual) within <class> </class> tags, and present the specific anomalous interval(s) within <answer> </answer> tags. This structured prompting strategy bridges low-level visual cues with high-level, human-interpretable reasoning in a unified framework. To illustrate this design, we present our full TSAD prompt in Fig. 7.

---

**Prompt for Anomaly Detection (Image Input)**

**Input Image**:

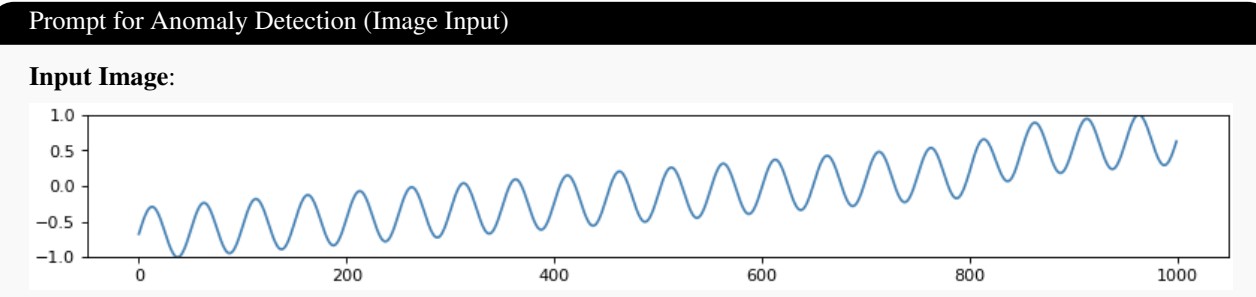

**Prompt**: You are a time series analysis expert. A time series plot of length 1000 is provided. Your task is to identify anomalous intervals along the x-axis.

The plot shows temporal patterns and typical value ranges based on both the x and y axes and may exhibit one type of anomaly from the set ["contextual point", "global point", "seasonal", "trend", "shapelet"]. Carefully examine the image to detect intervals that deviate from the expected behavior and infer the anomaly type.

Begin your detailed reasoning inside a <think>...</think> block. In this block, analyze the plot, identify suspicious intervals, and explain why they are considered anomalies and the type.

After reasoning, summarize your findings in the following format inside an <answer>...</answer> block: Each anomaly should be written as a [start, end] pair using x-axis positions. Then choose the type inside <class>...</class> from the set ["contextual point", "global point", "seasonal", "trend", "shapelet"].

Examples: <answer> [[120, 150], [320, 350]] </answer> <class>trend</class> or <answer> [[120, 150]] </answer> <class>global point</class> If no anomalies are found, return: <answer>[]</answer> <class>normal</class>

*Figure 7.* Prompt definition for time-series anomaly detection

## B.2. Details on ExpCoT.

We adopt the common anomaly taxonomy (Qiu et al., 2025) with five categories: (i) **Out-of-Range / Global Point**, (ii) **Contextual Point**, (iii) **Trend Shift**, (iv) **Seasonal/Frequency Deviation**, and (v) **Shapelet/Subsequence**. For each category, we pair characteristic signatures with classical, quantitatively verifiable analyses. ExpCoT is instantiated *per instance* from the ground-truth (GT) anomaly type and temporal annotation, and follows a disciplined three-stage path: OBSERVATION → REASONING & VALIDATION → CONCLUSION. In *Observation*, we perform a unified hierarchical scan of the series: starting with global distributions (e.g., extreme values), then examining structural properties (e.g., trend and periodicity), and finally analyzing localized patterns (e.g., subsequence dissimilarity) to surface candidate anomalies. *Reasoning & Validation* aligns the GT type and location with a targeted statistical probe and reports the resulting numerical evidence. *Conclusion* integrates these findings into a precise, GT-consistent statement of anomaly type and localization. Figures 9–11 illustrate some cases, and we provide the detailed pipeline for each anomaly type below.

**(i) Out-of-Range / Global Point.** OBSERVATION: Apply the defined global–structural–local scan to find salient deviations as candidates for anomaly detection. REASONING & VALIDATION: Apply a $k$-sigma envelope $[\mu - k\sigma, \mu + k\sigma]$ to formalize range departures; aggregate excursions into contiguous intervals and summarize $(\mu, \sigma)$ and the implied bounds. CONCLUSION: Retain the GT interval(s) as the definitive localization; envelope breaches serve as corroborating evidence.

**(ii) Contextual Point.** OBSERVATION: Apply the defined global–structural–local scan to find salient deviations as candidates for anomaly detection. REASONING & VALIDATION: Examine fixed-length, $z$-normalized subsequences using the Matrix Profile: let $d(i)$ be the discord distance and $i^* = \arg\max_i d(i)$. Standardize $\{d(i)\}$ to $z(i)$; if $z(i^*) > \tau$ (e.g., $\tau=3.5$), the subsequence $[i^*, i^*+m]$ constitutes strong evidence of a contextual departure. CONCLUSION: State the GT contextual-point interval(s) as final, summarizing the dominant discord and its standardized magnitude as quantitative support.

**(iii) Trend Shift.** OBSERVATION: Apply the defined global–structural–local scan to find salient deviations as candidates for anomaly detection. REASONING & VALIDATION: Smooth the series and analyze the gradient $g_t$; highlight segments where $|g_t - \bar{g}|$ exceeds a multiple of the empirical dispersion of $\{g_t\}$, and merge adjacent exceedances into candidate intervals indicating a shift in slope or level. CONCLUSION: Present the GT trend-shift span(s) as the conclusive localization,

together with the gradient summary (center, dispersion, and threshold) as supporting evidence.

**(iv) Seasonal/Frequency Deviation.** OBSERVATION: Apply the defined global–structural–local scan to find salient deviations as candidates for anomaly detection. REASONING & VALIDATION: Estimate the dominant period over sliding windows (FFT-based periodogram) and identify windows whose periods deviate beyond a robust tolerance around the typical period (e.g., median $\pm k \times 1.4826 \cdot$ MAD). Map these window-level deviations back to the time axis and merge them into intervals. CONCLUSION: Declare the GT seasonal/frequency interval(s) as final, reporting the typical period, its robust dispersion, and the deviation range as quantitative support.

**(v) Shapelet/Subsequence.** OBSERVATION: Apply the defined global–structural–local scan to find salient deviations as candidates for anomaly detection. REASONING & VALIDATION: Use a subsequence dissimilarity scan (e.g., Matrix Profile), prioritizing the most pronounced discord(s) and, when desired, assessing cross-scale stability across nearby window lengths to strengthen evidence. CONCLUSION: When GT specifies a shapelet/subsequence anomaly, return the GT interval(s) as the definitive localization and include the strongest dissimilar segment(s) as auxiliary evidence.

**Instantiation with Ground Truth.** For every instance, ExpCoT is generated from the GT class and temporal annotation: OBSERVATION anchors on the GT interval(s) and applies the unified scan (global $\rightarrow$ structural $\rightarrow$ local); REASONING & VALIDATION then selects the analysis matched to the GT type and reports concrete numerical evidence (global envelope deviation, standardized discord magnitude, smoothed-gradient exceedance, or dominant-period drift); CONCLUSION integrates these results and retains the GT interval(s) as the final localization, yielding a faithful, interpretable trace for supervising MLLM training. In practice, these traces are first generated automatically by code to provide quantified validation, and are subsequently refined by human experts for greater fluency and high-fidelity interpretability.

---

**Contextual Point Localization (GT [897,902])**

**Observation** — *Global Scan:* The series exhibits moderate variation with mean $\mu=0.123$ and standard deviation $\sigma=0.087$. The maximum standardized deviation $\max|z|=1.41$ suggests no extreme global outliers are present. *Structural Scan:* Gradient analysis reveals low trend instability ($\sigma_{\mathrm{grad}}=0.0034$), indicating stable baseline behavior. A clear dominant period appears at $\sim 47.2$ time units, revealing regular structural patterns. *Pattern Scan:* Discord search identifies the most dissimilar subsequence at $t \approx 899$ with discord score $=0.156$, suggesting potential local anomalies.
**Reasoning & Validation** — I observed that the subsequences within $[897, 902]$ are globally plausible but locally inconsistent, indicating a contextual anomaly. Matrix Profile analysis (window $m=50$) reveals strongest discord at $t \approx 899$ with $z$-score$=4.2$. Since $z > 3.5$, this confirms significant contextual deviation.
**Conclusion** — Integrating hierarchical scan insights with targeted contextual point validation, the evidence converges on anomalous behavior in interval $[897, 902]$. Therefore, the detected anomaly is classified as a contextual point, precisely localized to $[897, 902]$.

*Figure 8.* Example of ExpCoT reasoning trace for contextual point anomaly detection.

---

# C. Experimental Details

### C.1. Dataset Statistics

We evaluate three public resources to assess models' performance and generalizability across various TSAD scenarios. The detailed dataset statistics and anomaly coverage are summarized in Table 3.

1) **AnomLLM** (Zhou & Yu, 2024) provides controlled synthetic time-series anomaly detection benchmarks. Following the default generation settings, we generate eight anomaly types: out-of-range, point, frequency, trend, flat-trend, noisy-point, noisy-freq, and noisy-trend. They can be grouped into four categories: range, point, freq, and trend. For nomenclature consistency in this paper, we map the original task names to our taxonomy as follows: Range $\rightarrow$ Global point, Point $\rightarrow$ Contextual point, Freq $\rightarrow$ Seasonal, and Trend $\rightarrow$ Trend. Given this synthetic generation process, global (out-of-range) anomalies are typically the easiest to detect, whereas contextual point, trend, and seasonal anomalies are more difficult due to their reliance on local context, regime changes, and frequency shifts, respectively.

---

**Global Point Localization (GT [1245,1248])**

**Observation** — *Global Scan:* The series centers near zero with mean $\mu$=0.002 and moderate spread $\sigma$=0.098. However, the maximum standardized deviation $\max|z|$=6.73 indicates the presence of extreme values that deviate substantially from the typical range. *Structural Scan:* The series demonstrates high trend stability with minimal gradient variation ($\sigma_{\mathrm{grad}}$=0.0012), suggesting consistent baseline behavior. No dominant periodic structure is evident. *Pattern Scan:* Discord analysis identifies the most unusual subsequence at $t\approx$1246 with notable discord score =0.234, highlighting potential anomalous regions.

**Reasoning & Validation** — I observed that the values within $[1245, 1248]$ exhibit out-of-range behavior, representing clear global point anomalies. Applying $k$-$\sigma$ envelope $[-0.292, 0.296]$ validation: 4 points at $t \approx 1250$ exceed boundaries, confirming significant global deviation from normal range.

**Conclusion** — Integrating hierarchical scan insights with targeted global point validation, the evidence converges on anomalous behavior in interval $[1245, 1248]$. Therefore, the detected anomaly is classified as global point, precisely localized to $[1245, 1248]$.

*Figure 9.* ExpCoT reasoning trace for global point (out-of-range) anomaly detection.

---

**Trend Shift Localization (GT [8934,9078])**

**Observation** — *Global Scan:* The series shows slight negative bias with mean $\mu$=−0.045 and considerable variability $\sigma$=0.512. The moderate maximum standardized deviation $\max|z|$=2.18 suggests no extreme global outliers are present. *Structural Scan:* Gradient analysis reveals moderate trend instability ($\sigma_{\mathrm{grad}}$=0.0078), indicating some underlying directional changes. No dominant periodic structure is evident. *Pattern Scan:* Discord search locates the most dissimilar pattern at $t\approx$8967 with discord score =0.189, pointing to regions of structural change.

**Reasoning & Validation** — I observed that the segment within $[8934, 9078]$ shows a clear trend shift, with the long-term slope undergoing a marked change. Gradient analysis on smoothed series (window=21) shows baseline slope $\mu_{\mathrm{grad}}$=−0.0002 ± 0.0078. 47 points at $t \approx 8950$ exceed $k$-$\sigma$ gradient threshold (±0.0234), confirming abnormal trend changes.

**Conclusion** — Integrating hierarchical scan insights with targeted trend shift validation, the evidence converges on anomalous behavior in interval $[8934, 9078]$. Therefore, the detected anomaly is classified as trend shift, precisely localized to $[8934, 9078]$.

*Figure 10.* ExpCoT reasoning trace for trend shift anomaly detection.

2) **VisualTimeAnomaly** (Xu et al., 2025) converts numerical time series into images across various scenarios; in our study, we focus on the univariate setting and adhere to the default synthetic workflow. The benchmark includes point-wise (global/contextual) and range-wise (trend/seasonal/shapelet) anomalies for univariate series. Within this dataset, point-wise anomalies are the hardest to localize visually, whereas range-wise anomalies are comparatively easier due to their salient coarse-grained patterns.

3) **TSB-UAD** (Qiu et al., 2025) unifies 1,635 univariate series from the original TSB-UAD (Paparrizos et al., 2022) by filtering out low-quality series (e.g., those without anomalies or with an anomaly ratio >10%), resulting in a high-quality collection that includes both real-world and synthetic datasets. We adopt the official defaults and taxonomy. The TAB-UAD dataset covers both univariate and multivariate settings (treating each multivariate dataset as multiple univariate time series and evaluating them individually). The anomaly coverage includes point (global/contextual) and subsequence (trend/shapelet/seasonal) categories, as well as mixed types. The collected series span diverse domains such as industrial sensors, medical signals, finance, and web traffic, making the benchmark both comprehensive and representative of real-world anomaly detection challenges.

### C.2. Baselines

For each benchmark, we evaluate three groups of models. For the closed-source MLLMs, we access commercial APIs, including GPT-4o, GPT-4o-mini, Gemini-2.5-Pro, and Gemini-2.5-Flash-Lite. For the open-source counterparts, we rely on HuggingFace checkpoints such as Qwen/Qwen2.5-VL-72b-Instruct and its smaller variants (e.g., 32B/7B/3B). We further

---

**Frequency Deviation Localization (GT [2167,2189])**

**Observation** — *Global Scan:* The series maintains near-zero centering with mean $\mu$=0.014 and high variability $\sigma$=0.712. The moderate maximum standardized deviation $\max|z|$=1.89 indicates well-contained global fluctuations. *Structural Scan:* Gradient analysis shows moderate trend variation ($\sigma_{\text{grad}}$=0.0234), consistent with oscillatory behavior. A strong dominant period emerges at $\sim$50.0 time units, suggesting regular cyclical patterns. *Pattern Scan:* Discord analysis identifies the most atypical subsequence at $t$≈2178 with discord score =0.142, indicating potential frequency irregularities.

**Reasoning & Validation** — I observed that the oscillations within [2167, 2189] exhibit frequency deviation patterns, where the periodic structure changes relative to the expected seasonal baseline. Sliding period analysis ($W$=120) reveals typical period $\mu$=50.2 with robust $\sigma$=4.8. 8 windows fall outside robust range [35.8, 64.6], confirming frequency drift.

**Conclusion** — Integrating hierarchical scan insights with targeted frequency deviation validation, the evidence converges on anomalous behavior in interval [2167, 2189]. Therefore, the detected anomaly is classified as frequency deviation, precisely localized to [2167, 2189].

*Figure 11.* ExpCoT reasoning trace for frequency deviation anomaly detection.

*Table 3.* Comparison of AnomLLM, VisualTimeAnomaly and TSB-UAD.

| Category | AnomLLM | VisualTimeAnomaly | TSB-UAD |
|---|---|---|---|
| Synthetic | ✓ | ✓ | ✓ |
| Real-world | ✗ | ✓ | ✓ |
| Length range | 1,000 | ≈1,000 | 1,570–230,400 |
| Avg. anomaly rate | ~5% | ~5–15% | ~0.2–9.8% |
| **Anomaly Types** | | | |
| Global point | ✓ | ✓ | ✓ |
| Contextual point | ✓ | ✓ | ✓ |
| Trend | ✓ | ✓ | ✓ |
| Seasonal | ✓ | ✓ | ✓ |
| Shapelet | ✗ | ✓ | ✓ |

compare against supervised fine-tuned baselines, including Qwen2.5-VL-3B-SFT3.2k, fine-tuned on 3,200 instances, and Qwen2.5-VL-3B-SFT32k, fine-tuned on 32,000 instances.

In addition, we include two representative LLM-based temporal reasoning baselines. SIGLLM (Alnegheimish et al., 2024) is a GPT-3.5-based detector for anomaly identification. We evaluate SigLLM under the default settings provided in its official repository, using the original prompts and raw numerical inputs. TIMEMASTER (Zhang et al., 2025b), which builds on Qwen2.5-VL-3B with supervised fine-tuning (SFT) and GRPO and adopts image inputs, is also trained under its default public release. For all models except SIGLLM, we use the same prompt templates (see Figure 7) to ensure consistency and fairness.

### C.3. Metrics

We evaluate detection quality using the affiliation-based metrics introduced by Huet et al. (2022)[4]

**Setup.** Recall that a univariate time series of length $T$ is denoted by $\mathbf{X} = \{\mathbf{x}_t\}_{t=1}^{T}$. Ground-truth anomaly intervals are

---

[4]The official implementation of these metrics is publicly available, namely *Affi_Precision*, *Affi_Recall*, and their harmonic mean *Affi_F1*. These affiliation-based metrics can be viewed as event-level extensions of the classical precision/recall/F1-score to time-series anomaly detection (Huet et al., 2022). Affi_Precision and Affi_Recall evaluate each ground-truth event locally, and are parameter-free. Moreover, their construction via comparison to a random reference predictor makes the resulting scores both theoretically principled and practically useful for TSAD, especially in LLM-based TSAD settings (Zhou & Yu, 2024; Liu et al., 2024; Xu et al., 2025). Below, we provide their detailed definitions.

given by

$$\mathcal{A} = \big\{(t_s^{(i)}, t_e^{(i)})\big\}_{i=1}^{k}, \qquad 1 \le t_s^{(i)} \le t_e^{(i)} \le T,$$

where each interval $(t_s^{(i)}, t_e^{(i)})$ denotes the $i$-th anomalous segment (with $t_s^{(i)} = t_e^{(i)}$ corresponding to a single-point anomaly). We assume these intervals are pairwise disjoint. For convenience, we identify each interval with the corresponding set of time indices,

$$J_i = \{t \in \{1, \ldots, T\} : t_s^{(i)} \le t \le t_e^{(i)}\}.$$

Thus the collection of ground-truth events is

$$\mathcal{J} = \{J_j\}_{j=1}^{n},$$

where $n = k$ and the $J_j$ are pairwise disjoint subsets of $\{1, \ldots, T\}$.

Similarly, we denote the predicted anomaly intervals by

$$\widehat{\mathcal{A}} = \big\{(\hat{t}_s^{(i)}, \hat{t}_e^{(i)})\big\}_{i=1}^{\hat{k}},$$

and write

$$\widehat{J}_i = \{t \in \{1, \ldots, T\} : \hat{t}_s^{(i)} \le t \le \hat{t}_e^{(i)}\}, \qquad \widehat{\mathcal{J}} = \{\widehat{J}_i\}_{i=1}^{\hat{k}}.$$

All sets $J_j$ and $\widehat{J}_i$ are subsets of the index set $\mathcal{T} = \{1, \ldots, T\}$. For any $A \subseteq \mathcal{T}$, we write $|A|$ for its cardinality. For $t \in \mathcal{T}$ and $Y \subseteq \mathcal{T}$, we define

$$\mathrm{dist}(t, Y) = \min_{y \in Y} |t - y|$$

as the distance (in time indices) from $t$ to the set $Y$, with the convention that $\mathrm{dist}(t, \varnothing) = +\infty$.

**Affiliation regions.** Following (Huet et al., 2022), we partition the time index set $\mathcal{T}$ into *affiliation regions* $\{E_j\}_{j=1}^{n}$, one for each ground-truth event $J_j$:

$$E_j = \big\{t \in \mathcal{T} : j = \arg \min_{k \in \{1, \ldots, n\}} \mathrm{dist}(t, J_k)\big\},$$

with ties broken arbitrarily so that $\{E_j\}_{j=1}^{n}$ forms a partition of $\mathcal{T}$, i.e. $\mathcal{T} = \biguplus_{j=1}^{n} E_j$ and $E_j \cap E_k = \varnothing$ for $j \ne k$. For each $j$, we denote by

$$\widetilde{P}_j = \Big(\bigcup_{i=1}^{\hat{k}} \widehat{J}_i\Big) \cap E_j$$

the subset of predicted anomalous time indices that fall inside the affiliation region $E_j$.

**Random reference predictor.** For each $j \in \{1, \ldots, n\}$, we define a random reference predictor by drawing a time index

$$X_j \sim \mathrm{Unif}(E_j),$$

uniformly at random from $E_j$. The *precision-side baseline distance* is

$$D_j^{\mathrm{prec}} = \mathrm{dist}(X_j, J_j),$$

and its survival function (complementary CDF) is

$$\overline{F}_j^{\mathrm{prec}}(d) = \mathbb{P}\big(D_j^{\mathrm{prec}} \ge d\big), \qquad d \ge 0.$$

Intuitively, $\overline{F}_j^{\mathrm{prec}}(d)$ measures how likely a random prediction in $E_j$ lies at distance at least $d$ from the true event $J_j$.

For the recall side, for each $j$ and each time index $t \in J_j$, we define

$$D_{j,t}^{\mathrm{rec}} = \mathrm{dist}(t, X_j),$$

and the corresponding survival function

$$\overline{F}_{j,t}^{\mathrm{rec}}(d) = \mathbb{P}\big(D_{j,t}^{\mathrm{rec}} \ge d\big), \qquad d \ge 0.$$

**Affi_Precision.** For a fixed ground-truth event $J_j$, the *local affiliation-precision score* $P_{\text{prec}}(j)$ compares the actual predictions in $E_j$ to the random baseline:

$$
P_{\text{prec}}(j) = \begin{cases} \dfrac{1}{|\widetilde{P}_j|} \sum_{t \in \widetilde{P}_j} \overline{F}_j^{\text{prec}}\big(\text{dist}(t, J_j)\big), & \text{if } |\widetilde{P}_j| > 0, \\ (\text{ignored}), & \text{if } |\widetilde{P}_j| = 0. \end{cases}
$$

Only those events with $|\widetilde{P}_j| > 0$ contribute to the global precision. Let

$$
S = \big\{ j \in \{1, \ldots, n\} : |\widetilde{P}_j| > 0 \big\}
$$

be the set of ground-truth events for which at least some prediction mass falls into $E_j$. The global *Affi_Precision* is defined as

$$
\text{Affi\_Precision} = \begin{cases} \dfrac{1}{|S|} \sum_{j \in S} P_{\text{prec}}(j), & \text{if } |S| > 0, \\ 0, & \text{if } |S| = 0. \end{cases}
$$

**Affi_Recall.** For the recall side, each ground-truth event $J_j$ defines a local score $P_{\text{rec}}(j)$ by averaging, over all time indices $t \in J_j$, how much better the prediction $\widetilde{P}_j$ is than the random baseline:

$$
P_{\text{rec}}(j) = \frac{1}{|J_j|} \sum_{t \in J_j} \overline{F}_{j,t}^{\text{rec}}\big(\text{dist}(t, \widetilde{P}_j)\big),
$$

where

$$
\text{dist}(t, \widetilde{P}_j) = \min_{z \in \widetilde{P}_j} |t - z|,
$$

with the convention that if $\widetilde{P}_j = \varnothing$, then $\text{dist}(t, \widetilde{P}_j) = +\infty$ and $\overline{F}_{j,t}^{\text{rec}}(\text{dist}(t, \widetilde{P}_j)) = 0$.

The global *Affi_Recall* is obtained by averaging $P_{\text{rec}}(j)$ over all ground-truth events:

$$
\text{Affi\_Recall} = \frac{1}{n} \sum_{j=1}^{n} P_{\text{rec}}(j).
$$

**Affi_F1.** Finally, the *Affi_F1* score is defined as the harmonic mean of Affi_Precision and Affi_Recall. Let

$$
P = \text{Affi\_Precision}, \qquad R = \text{Affi\_Recall},
$$

then

$$
\text{Affi\_F1} = \begin{cases} 0, & \text{if } P + R = 0, \\ \dfrac{2PR}{P + R}, & \text{otherwise.} \end{cases}
$$

By construction, Affi_Precision, Affi_Recall, and Affi_F1 all take values in the interval $[0, 1]$.

### C.4. Implementation Details

**Time-Series Image Input.** We follow the common plotting conventions used in prior work on MLLMs (Xu et al., 2025; Zhou & Yu, 2024; Zhang et al., 2025b) to ensure fairness. The line plots do not include shaded or highlighted regions, and anomalous intervals are not explicitly marked. Each time-series image is rendered at a resolution of $805 \times 124$ pixels.

**Training Setup.** We initialize our backbone with the publicly available Qwen2.5-VL-3B-Instruct and Qwen2.5-VL-7B-Instruct checkpoints. Our overall training pipeline only includes a TimerPO stage based purely on reinforcement learning. We build our implementation on the public RL training library and the temporal reasoning training framework. We summarize our hyperparameter settings in Table 4, where the GRPO configuration follows TIMEMASTER for fairness. The models are trained on 3,200 synthetic samples from ANOMLLM and evaluated on the ANOMLLM synthetic test set, VISUALTIMEANOMALY, and TSB-UAD, which cover broader anomaly types and varying sequence lengths to assess generalization to unseen real-world scenarios.

*Table 4.* Hyperparameter settings.

| Algorithm | Hyperparameter | Value | Algorithm | Hyperparameter | Value |
|---|---|---|---|---|---|
| GRPO | Max response length | 1024 | TimerPO | $\lambda^{\text{fmt}}$ | 0.1 |
| | Batch size | 128 | | $\lambda^{\text{cls}}$ | 0.2 |
| | Mini-batch size | 128 | | $\lambda^{\text{loc}}$ | 0.7 |
| | KL loss coefficient | 0.001 | | $\alpha$ | 0.3 |
| | Group size | 5 | | | |
| | Learning rate | 1e-6 | | | |

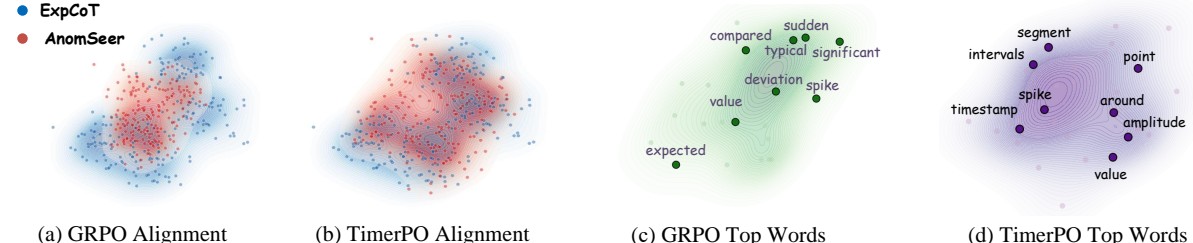

    (a) GRPO Alignment    (b) TimerPO Alignment    (c) GRPO Top Words    (d) TimerPO Top Words

*Figure 12.* Comparison of distributional alignment between ExpCoT (blue) and ANOMSEER (red) outputs, along with token usage under GRPO and TimerPO training.

## C.5. System Configuration

All experiments were conducted on a computing setup equipped with 4 NVIDIA A100-SXM4 GPUs (80 GB each) and 4 NVIDIA RTX A6000 GPUs (48 GB each) for Qwen-3B, and 4 NVIDIA H100-SXM4 GPUs (96 GB each) for Qwen-7B.

## D. More Experimental Results

### D.1. Confidence Intervals and Computational Cost

To complement the results in the main paper, we provide the complete set of performance metrics corresponding to Table 1, including mean values over three runs together with their 95% confidence intervals. As shown in Table 5, the consistently small intervals support the robustness of our findings and indicate that ANOMSEER performs stably across repeated trials.

We also report the computational profile of ANOMSEER (3B) trained on NVIDIA RTX A6000 GPUs with 48 GB of memory. The training phase requires 12.4 hours of wall-clock time using four GPUs in parallel. For inference, the model operates on a single GPU, utilizing approximately 7 GB of memory and achieving an average latency of 4.8 seconds per time-series sample. These computational characteristics fall within acceptable limits for practical deployment in TSAD scenarios.

### D.2. Learning Curves & Data Scaling

We present the learning curves and data-scaling results in Figure 13. We first observe that the learning curves for both the 3B and 7B models exhibit stable and monotonic improvement, with performance rising rapidly during the initial 50-100 training steps before gradually stabilizing. In addition, scaling the training set from 1k to 5k examples yields consistent gains across all four tasks. The average Affinity F1 score continues to improve as the dataset grows, with no clear signs of saturation. These results suggest that the current data regime remains in a growth phase, and further increasing the amount of training data is likely to yield additional performance gains.

### D.3. Optimization and Alignment Ablation

**Advantage-level orthogonalization vs. gradient-level projection.** We compared TimerPO to two multi-objective optimization baselines: (i) a weighted-sum objective (no projection) and (ii) PCGrad-style gradient orthogonalization (Yu et al., 2020). Table 6 summarizes the results. TimerPO consistently outperforms both weighted-sum and gradient-level projection across all anomaly types. Orthogonalizing auxiliary signals at the advantage level promotes complementary contributions

*Table 5.* Mean $\pm$ 95% confidence interval half-width over 3 seeds.

| Method | Classification Accuracy | Location | | | | | | | | | | | | | | Avg F1 |
|---|---|---|---|---|---|---|---|---|---|---|---|---|---|---|---|---|
| | | Frequency | | | Trend | | | Range | | | Point | | | | |
| | | P | R | F1 | P | R | F1 | P | R | F1 | P | R | F1 | | |
| TimeMaster-3B | 57.90±1.49 | 57.30±1.24 | 50.30±0.25 | 51.40±0.50 | 76.00±1.24 | 77.30±0.25 | 76.60±1.24 | 77.80±1.24 | 83.50±0.25 | 80.10±0.25 | 77.70±1.24 | 82.10±0.25 | 79.60±1.24 | | 71.92±0.81 |
| ANOMSEER-3B | 62.80±1.24 | 63.70±1.24 | 58.40±1.24 | 58.90±1.24 | 84.20±0.50 | 85.90±0.25 | 84.90±0.25 | 83.30±0.75 | 89.20±0.25 | 85.60±0.25 | 86.00±0.25 | 90.30±0.25 | 87.80±0.25 | | 79.30±0.50 |
| ANOMSEER-7B | 65.00±1.24 | 68.30±1.24 | 59.40±0.50 | 60.80±0.50 | 86.60±1.24 | 89.00±1.24 | 87.70±1.24 | 91.60±0.25 | 97.80±0.99 | 94.30±0.25 | 93.40±0.25 | 96.90±1.24 | 94.90±0.25 | | 84.42±0.56 |

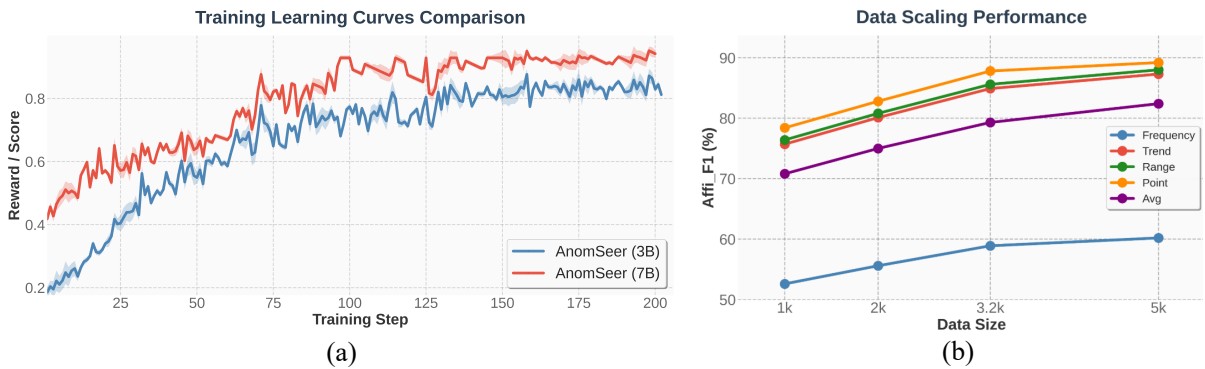

*Figure 13.* (a) Learning curves of training score versus training steps for the 3B and 7B models, and (b) data-scaling performance for the 3B model evaluated from 1k to 5k training examples.

prior to gradient computation, whereas PCGrad only modifies gradients when explicit conflicts are detected. This reduces partial interference between objectives and results in smoother, lower-variance optimization trajectories.

*Table 6.* Comparison of orthogonalization strategies (top) and alignment objectives (bottom).

| Method | Freq. | Trend | Range | Point | Avg |
|---|---|---|---|---|---|
| *Orthogonalization strategies* | | | | | |
| TimerPO (ours) | 58.9 | 84.9 | 85.6 | 87.8 | **79.3** |
| Weighted-sum (no proj.) | 53.5 | 81.1 | 83.5 | 85.4 | 75.9 |
| PCGrad (gradient level) | 54.2 | 80.2 | 84.5 | 86.4 | 76.3 |
| *Alignment objectives* | | | | | |
| TimerPO (ours) | 58.9 | 84.9 | 85.6 | 87.8 | **79.3** |
| Cosine similarity | 42.2 | 73.8 | 84.1 | 86.8 | 71.7 |
| CLIP-style similarity | 48.5 | 74.1 | 84.3 | 86.7 | 73.4 |

**Replacing OT with cosine or contrastive similarity.** To isolate the contribution of OT-based alignment, the OT module in TimerPO was replaced with two alternatives: (i) token-wise cosine similarity and (ii) a CLIP-style InfoNCE objective (temperature $= 0.07$). As shown in Table 6, OT yields a 6.8% improvement in average F1 relative to cosine and contrastive similarity. OT provides structure-aware alignment by modeling semantic distances between reasoning tokens rather than treating tokens independently. These findings indicate that OT geometry plays an essential role in aligning model reasoning with temporally structured anomaly patterns.

## D.4. Extensibility

We now discuss the extensibility of the proposed method, with results summarized in Tab. 7. **1) Multivariate time series.** Although the main experiments focus on univariate data, the framework is not limited to this setting. Multivariate inputs can be converted into a unified visual representation by rendering each variable as a subplot within a single image. Empirical results on a multivariate benchmark demonstrate that the method generalizes effectively beyond the univariate setting. **2) Short-term and boundary anomalies.** Short-duration or boundary anomalies are typically underrepresented in existing datasets and therefore challenging to detect reliably. A simple targeted augmentation strategy yields notable improvements on a dedicated evaluation set of such cases. These findings indicate that lightweight preprocessing and sampling adjustments can enhance robustness in challenging anomaly scenarios.

*Table 7.* Results for multivariate evaluation (top) and short/boundary anomaly robustness (bottom).

| Setting | GPT-4o | Gemini-2.5 | Qwen2.5-VL | Ours (7B) |
|---|---|---|---|---|
| Multivariate (synthetic / real-world) | 62.7 / 54.2 | 77.3 / 65.2 | 45.0 / 24.5 | **83.5 / 72.4** |
| Short / boundary anomalies | Original: 56.2 | | **+ Augmentation: 75.8** | |

### D.5. Comparison with Traditional TSAD Methods

To provide a unified view of classical time-series anomaly detection (TSAD) methods and our framework, Table 8 summarizes representative baselines across four anomaly types. Traditional approaches such as FFT, Matrix Profile, gradient-based detection, ARIMA, and statistical thresholding operate directly on raw signals and typically produce detection outputs only. While they can perform well in specific scenarios, they often rely on careful parameter tuning (e.g., window selection or differencing) and exhibit limited robustness across diverse anomaly patterns.

In contrast, ANOMSEER outperforms traditional approaches across all anomaly types and, more importantly, supports detection, classification, and natural language reasoning within a single model. This broader output capability enables interpretability and generalization across diverse anomaly patterns, rather than optimizing for a single metric or domain-specific signal property.

*Table 8.* Comparison with classical TSAD baselines and the proposed ANOMSEER.

| Method | Capability | Freq. | Trend | Range | Point | Avg |
|---|---|---|---|---|---|---|
| FFT | Location only | 65.9 | 18.0 | 28.5 | 27.8 | 35.1 |
| Matrix Profile | Location only | 11.4 | 29.4 | 67.2 | 87.4 | 48.9 |
| Gradient | Location only | 57.1 | 58.5 | 55.8 | 65.9 | 59.3 |
| Ensemble (voting) | Location only | 59.0 | 17.4 | 69.2 | 92.4 | 59.5 |
| ARIMA | Location only | 62.6 | 4.9 | 67.3 | 74.2 | 52.3 |
| Thresholding | Location only | 57.6 | 25.2 | 60.0 | 47.6 | 47.6 |
| ANOMSEER (3B) | Loc. + Cls. + Reasoning | 58.9 | 84.9 | 85.6 | 87.8 | **79.3** |
| ANOMSEER (7B) | Loc. + Cls. + Reasoning | 60.8 | 87.7 | 94.3 | 94.9 | **84.4** |

### D.6. Details on Effect of TimerPO

To highlight the advantage of TimerPO over vanilla GRPO in temporal reasoning, we further compare their behaviors in Figure 12. While GRPO narrows the distributional gap to some extent, the model outputs remain relatively constrained and still exhibit a clear mismatch compared to expert reasoning. A similar pattern is evident in token usage: GRPO-trained outputs are dominated by outcome-oriented words such as compared and expected, whereas TimerPO encourages the use of more fine-grained, temporally grounded terms like timestamp, intervals, and amplitude, which anchor reasoning to concrete temporal structures. These findings confirm that TimerPO provides a more principled enhancement over GRPO, enabling models to move beyond surface outcome alignment toward genuine temporal reasoning.

### D.7. More Case Studies on Reasoning

We provide several case studies illustrating our model's complete reasoning process on corresponding data visualizations. These examples show our approach's effectiveness in focusing on specific segments and timestamps for fine-grained analysis. We also present a failure case in Fig. 14: a short anomaly within the interval [998, 1000] at the sequence's very end goes undetected by ANOMSEER, which incorrectly classifies it as 'normal'. This highlights the need for future work to improve the sensitivity of MLLMs to such boundary-case anomalies.

## E. Extended Related Work

**LLM-based time series anomaly detection (TSAD)** is an emerging area, with several exploratory methods recently proposed. These approaches vary in modalities, backbones, and integration strategies. For example, SigLLM (Alnegheimish et al., 2024) and CoLLaTe (Chen et al., 2025) use numeric–text inputs with GPT-3.5 or GPT-4, relying on prompting with external post-processing or task-specific TSAD modules, but without reasoning ability. LLMAD (Liu et al., 2025a)

Response of ANOMSEER on the ground-truth trend anomaly [[800, 850]].

**Input Image**:

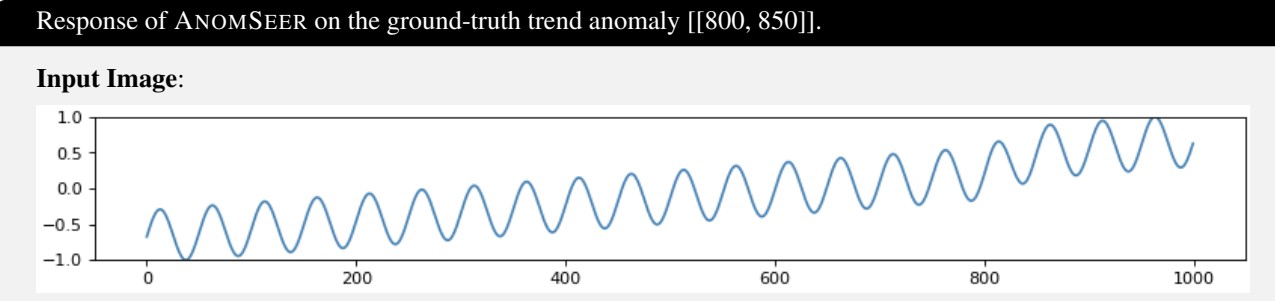

**ANOMSEER Response**: <think> The plot shows a general wave-like pattern with intervals where the amplitude becomes slightly elevated. Such behavior could hint at a potential trend change, but the deviations are not very pronounced. The global structure remains smooth, with no sudden spikes or sharp dips that would qualify as clear anomalies. However, I did notice a small irregularity forming around timestep 850, where the local pattern begins to diverge slightly from the surrounding trend. The fluctuation is modest in amplitude, with peaks rising only marginally higher and troughs dipping slightly deeper than the preceding cycles. This subtle change suggests a weak signal of trend variation rather than a strong anomaly. </think>
<answer> [[800, 900]] </answer>
<class> trend </class>

---

Response of ANOMSEER on the contextual point anomaly [[200,231],[622,673]]

**Input Image**:

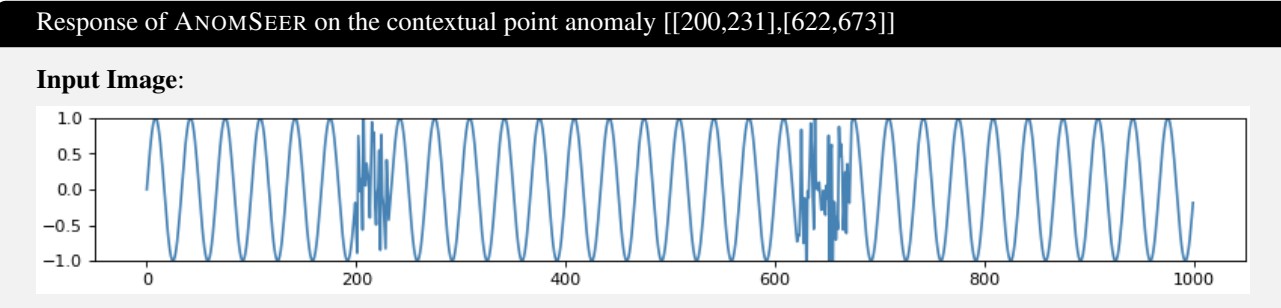

**ANOMSEER Response**: <think> The time series exhibits two noticeable intervals that deviate from the surrounding pattern. The first occurs between roughly 200 and 250, while the second spans around 600 to 700. Both segments rise sharply compared with the smoother baseline, resembling local spikes rather than gradual changes. These departures stand out in their immediate neighborhoods but do not exceed the overall global range of the series. Because these deviations break the contextual continuity of the waveform, they are better characterized as contextual point anomalies. </think>
<answer> [[200, 250], [600, 700]] </answer>
<class> contextual point </class>

---

augments GPT-4-turbo with retrieval-based domain knowledge to support anomaly classification and localization, though it requires an external database for prompting. On the multimodal side, TAMA (Zhuang et al., 2024) and VLM4TS (He et al., 2025) employ image–text inputs with GPT-4o, together with post-processing or ViT-based components. More recently, Time-RA (Yang et al., 2025) applies large-scale SFT on Qwen2.5-VL-7B, but its coverage remains incomplete, particularly in anomaly localization. In contrast, our method uses a compact open-source backbone, Qwen2.5-VL-3B/7B, and reinforcement learning to directly equip the model with anomaly classification, localization, and reasoning, without external modules or proprietary APIs.

**Optimal Transport** (OT) offers a principled geometric framework for aligning probability distributions and has seen increasing adoption in both reinforcement learning (RL) and large language model (LLM) alignment. In RL, OT has been leveraged to structure learning signals and align task or policy distributions (Klink et al., 2022; Asadulaev et al., 2024; Chen et al., 2020), notably in curriculum design (Klink et al., 2022) and as a regularizer for offline policy learning (Asadulaev et al., 2024). Within LLM, OT has been used to support preference modeling (Li et al., 2025a; Melnyk et al., 2024; Désidéri, 2012;

Response of ANOMSEER on the global point anomaly [[358,380],[910,931]]

**Input Image**:

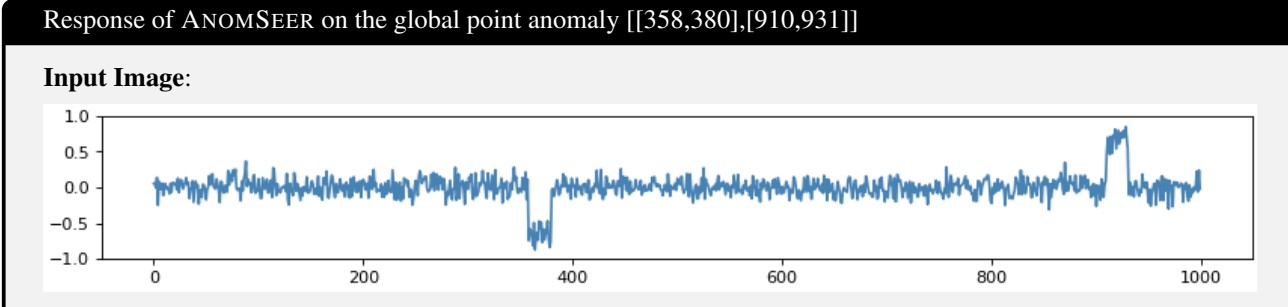

**ANOMSEER Response**: <think> The plot contains several sharp, discontinuous excursions in amplitude. By scanning the sequence for abrupt changes that break this continuity, I identified three distinct intervals: one near 350, another around 880, and a third close to 900. Each spike pops outside the local band, then quickly snaps back. These intervals deviate noticeably from the surrounding baseline, suggesting global point anomalies in the series. </think>
<answer> [[350, 450], [880, 950]] </answer>
<class> global point </class>

Response of ANOMSEER on the freq anomaly [[653,676],[811,846]]

**Input Image**:

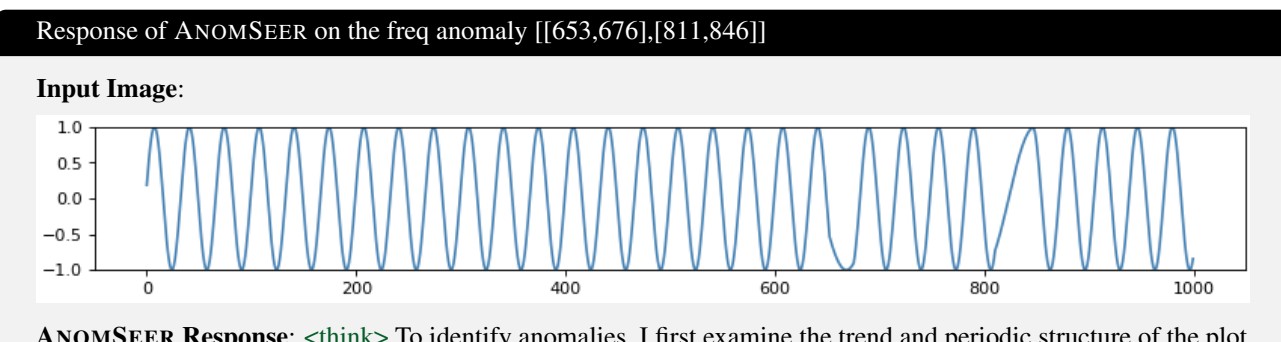

**ANOMSEER Response**: <think> To identify anomalies, I first examine the trend and periodic structure of the plot. The early section displays a regular sinusoidal rhythm with no unusual deviations. Around the 650th timestep, however, the frequency begins to distort, showing dips and irregular oscillations that break the expected cycle. In addition, I observed another disturbance near the 850th mark, where the pattern again deviates from the baseline. This suggests a change in the underlying dynamics, with the presence of a seasonal anomaly disrupting the expected pattern. </think>
<answer> [[580, 700], [850, 900]] </answer>
<class> seasonal </class>

Xu et al., 2026; Li et al., 2025b), by aligning full reward distributions (Melnyk et al., 2024) or applying token-level weighting schemes to highlight semantically important regions (Li et al., 2025a). Most of these approaches focus on final outcome alignment, operating over entire sequences or aggregated behaviors. But our work applies OT at the reasoning-token level, aligning the model's intermediate reasoning steps with structured ExpCoT traces derived from classical TSAD primitives. This enables process-level supervision, enhancing the model's temporal reasoning capabilities rather than merely refining output preferences.

**Multi-objective optimization** methods (Désidéri, 2012; Yu et al., 2020; Liu et al., 2021; Wei & Hu, 2024) aim to stabilize training across competing tasks by projecting conflicting gradients into compatible directions. For example, PCGrad (Yu et al., 2020) explicitly projects one task's gradient onto the normal plane of another when conflicts arise. In contrast, our TimerPO introduces orthogonal projection in the advantage space, not to resolve inter-task interference, but to preserve the independent contribution of an auxiliary reasoning advantage. Since this auxiliary signal reflects structured supervision rather than a separate objective, our projection design allows it to complement the main anomaly detection reward without disruption. To the best of our knowledge, this is the first approach to combine token-level OT alignment with advantage-space disentanglement to enhance temporal reasoning in multimodal LLMs.

Failure case where ANOMSEER fails to detect the ground-truth trend anomaly at [998,1000]

**Input Image**:

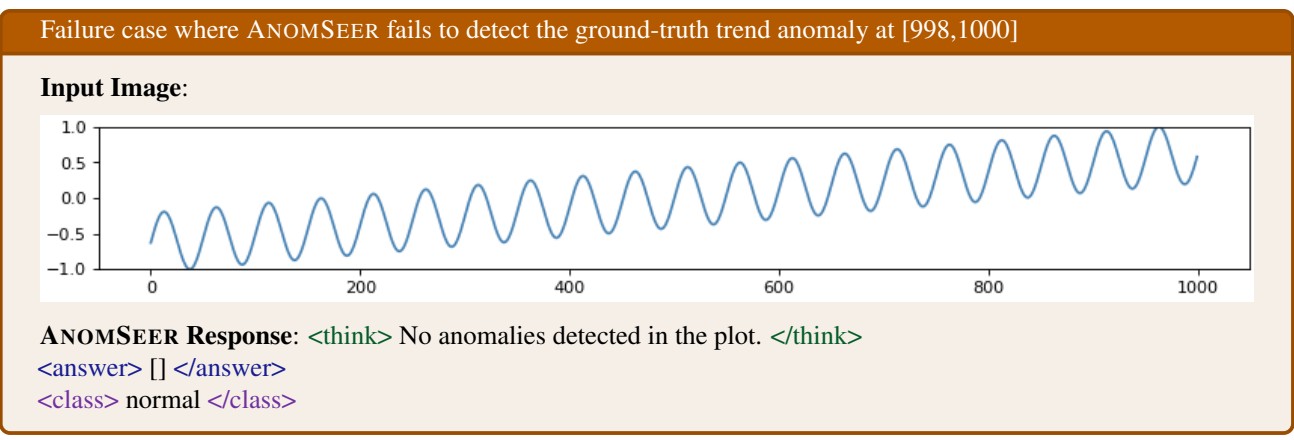

**ANOMSEER Response**: <think> No anomalies detected in the plot. </think>
<answer> [] </answer>
<class> normal </class>

*Figure 14.* Failure case

