# OpenReview forum: "AnomSeer: Reinforcing Multimodal LLMs to Reason for Time-Series Anomaly Detection"
_ICML.cc/2026/Conference — ICML 2026 regular_

### Official Review · Reviewer_w2nX · 2026-02-28

**Soundness:** 3
**Presentation:** 4
**Significance:** 4
**Originality:** 3
**Overall Recommendation:** 5
**Confidence:** 4

**Summary:**

This paper propose AnomSeer to tackle the difficulty of MLLMs in time‑series anomaly detection, where coarse heuristics hinder detailed reasoning. The method generates expert chain‑of‑thought traces that ground the model’s decisions in fine‑grained classical analyses such as statistical metrics and frequency transforms, unifying anomaly classification, localization, and explanation. It also introduces TimerPO, a time‑series‑grounded policy‑optimization framework that adds a transport‑based advantage term and an orthogonal projection to preserve the primary detection objective while leveraging the auxiliary granular signal. Evaluations across varied anomaly types show that the proposed method surpasses larger commercial baselines in both classification and localization accuracy, and yields coherent reasoning traces that validate its predictions.

**Compliance With Llm Reviewing Policy:**

Affirmed.

**Key Questions For Authors:**

- Section 4.2, “Time‑Series Reasoning Advantage”: I am curious about the robustness of the cost matrix C in aligning the model’s reasoning with the ExpCoT distribution. The underlying assumption of the cost matrix is that the order of reasoning blocks in each trace is aligned. Please explain how practical this is in real situations.

**Limitations:**

The paper has a section on impact statement.

**Strengths And Weaknesses:**

The proposed techniques enable multimodal LLMs to detect and reason about time‑series anomalies in a fine‑grained and accurate manner. This work can bring significant value and impact to the time‑series community by bridging the gap between the latest advancements in LLM‑based reasoning and traditional time‑series analysis. The paper is well written and clearly describes each step of the proposed method.

One comment is that the reasoning process described in the paper relies solely on patterns extracted from the time‑series data, such as repeated waveforms, spikes, and similar characteristics. In real‑world analysis, reasoning often extends beyond the properties of the data and incorporates domain‑specific knowledge and patterns. The authors are encouraged to extend this work to integrate domain‑specific information into the reasoning process, as such an extension would significantly boost the practical value of the approach in real‑world applications.

I also suggest that the authors conduct a study to illustrate how many ground‑truth annotations are required to achieve the expected performance.  In real applications, how difficult to obtain those annotations?

---

> ### Author Rebuttal · Authors · 2026-03-31
>
> We thank you for the encouraging feedback and for recognizing the significant value and impact AnomSeer brings to the time-series community. Below we address your comments.
>
> ### Comment 1: Integrating domain-specific knowledge
>
> We fully agree that real-world anomaly reasoning often goes beyond patterns in the time series itself and may depend on **dataset background** and **source context**, such as metadata about the data collection environment. One practical advantage of AnomSeer is that this type of information can be incorporated directly through the **context prompt $c$**, without changing the model architecture.
>
> As suggested, we conducted an additional study on two real-world datasets by adding brief **dataset-level background information** from the dataset descriptions to the prompt: Yahoo (real production traffic) and SMAP (NASA rover telemetry and sensor data). With this additional context, the model achieved **improved detection performance**:
>
> | Dataset | Base AnomSeer | AnomSeer + domain context |
> | --- | --- | --- |
> | Yahoo | 88.2 | 89.7 |
> | SMAP | 66.9 | 68.4 |
>
> These results suggest that AnomSeer can effectively **leverage domain-specific information** when it is available, improving its **practical value in real-world applications**. We appreciate this suggestion, and it motivates us to further explore the integration of richer contextual knowledge into time-series anomaly reasoning as a promising future direction.
>
> ### Comment 2: Annotation efficiency and annotations cost
>
> We thank the reviewer for raising this important practical question. Regarding how many ground-truth annotations are needed, our data scaling analysis in Appendix E.2 (Figure 13b) shows that performance improves rapidly and becomes largely stable with only **1k–3k** training samples. In our main setting, we use **3,200** training instances.
>
> Regarding annotation difficulty and cost in real applications, a key advantage of our pipeline is that the training instances are derived from the synthetic AnomLLM dataset through an almost fully automatic ExpCoT construction workflow. As a result, the reasoning traces and anomaly annotations used for training do not require large-scale human labeling, making the annotation cost effectively **negligible**.
>
> Importantly, although trained only on synthetic traces, AnomSeer still generalizes well in a zero-shot manner to real-world datasets such as TSB-UAD. This suggests that strong performance does not depend on expensive manual reasoning annotations, which improves the practical applicability of the approach in real-world settings.
>
> ### Q1: Robustness of the cost matrix $C$ and the alignment assumption
>
> Thank you for this question. We would like to clarify that the cost matrix $C$ does **not** require **strict alignment** between reasoning blocks. One reason we use Optimal Transport (OT), instead of sequence-level objectives such as cross-entropy, is precisely that **OT does not assume exact positional matching**.
>
> Specifically, $C$ is built from pairwise semantic cosine distances between any token $n$ in the model-generated trace and any token $m$ in the ExpCoT trace, **regardless of their absolute positions**. Based on this matrix, the entropic-regularized Sinkhorn algorithm computes a **soft many-to-many transport plan** that minimizes the overall semantic transport cost [1] [2]. As a result, the alignment is based on **semantic correspondence rather than token order**. Even when the model reorders reasoning components or uses different phrasing, OT can still match related content through the transport plan. This makes TimerPO more robust to the structural flexibility of LLM-generated reasoning than strict token-level matching objectives.
>
> [1] From Word Embeddings To Document Distances. ICML'15
>
> [2] Sinkhorn Distances: Lightspeed Computation of Optimal Transport. NeurIPS'13

---

> > ### Author Rebuttal · Reviewer_w2nX · 2026-04-06
> >
> > I am satisfied with the authors' rebuttal.

---

> > > ### Author Response · Authors · 2026-04-06
> > >
> > > Thank you again for your insightful review. We sincerely appreciate your continued engagement and thoughtful suggestions, which have helped improve the quality of our work. We are grateful for your time and support.

---

### Official Review · Reviewer_bmp3 · 2026-03-11

**Soundness:** 2
**Presentation:** 3
**Significance:** 2
**Originality:** 3
**Overall Recommendation:** 3
**Confidence:** 4

**Summary:**

This paper presents a novel time series anomaly detection method called ANOMSEER. Its core lies in the introduction of ExpCoT and TimerPO, which generate verifiable fine-grained reasoning traces based on classical analysis, and the introduction of time series advantage signals and orthogonal projection mechanisms based on optimal transmission, aiming to enable the model to provide refined reasoning while generating detection results. Experiments show that ANOMSEER based on Qwen2.5-VL-3B/7B-Instruct performs outstandingly in various anomaly scenarios.

**Compliance With Llm Reviewing Policy:**

Affirmed.

**Key Questions For Authors:**

(1)Does the ANOMSEER generalize effectively to dynamic or non-stationary time series environments?
(2)To what extent can it detect previously unseen anomaly patterns?
(3)Does splitting the time series data into tokens affect accuracy when dealing with long sequences? What if some anomalies or data characteristics are very local information, or appear between tokens?
(4)To what extent does model accuracy degrade under resource-constrained deployment conditions. Specifically, when hardware capabilities fall below the recommended threshold?

**Limitations:**

ANOMSEER focuses on univariate time-series data. However, in real industrial scenarios, it is often multivariate time series data anomaly detection, which limits the use of this method in real scenarios. And this model relies on high computing power. This significantly increases the deployment cost.

**Strengths And Weaknesses:**

Regarding methodological soundness, the expert chain-of-thought employs statistical diagnostic techniques to generate structured, domain-expert–informed reasoning trajectories from real-world time series data. However, several critical questions remain considering: (1) Does the ANOMSEER generalize effectively to dynamic or non-stationary time series environments? (2) To what extent can it detect previously unseen anomaly patterns? (3) Given that the ANOMSEER was trained exclusively on the AnomLLM dataset, how well does this benchmark reflect the diversity, severity, and contextual complexity of real-world anomalous phenomena?
In terms of presentation, the paper has a well-structured format, detailed descriptions of hyperparameters, and provides complete pseudo-code, which facilitates reproducibility.
Regarding significance, this method primarily addresses the modality alignment challenge between MLLMs and TSAD tasks. However, its computational requirements are relatively high. A key practical concern therefore arises: to what extent does model accuracy degrade under resource-constrained deployment conditions. Specifically, when hardware capabilities fall below the recommended threshold?
In terms of originality, the paper transformed the traditional time series anomaly detection problem into a "generative reasoning" task, and demonstrated certain innovation in the design of prompts as shown in Fig. 7.

---

> ### Author Rebuttal · Authors · 2026-03-31
>
> We sincerely thank your comments and address the questions below.
> ### Q1 & W1: Generalization to dynamic/non-stationary environments
> * We evaluated *cross-distribution* generalization in Sec. 5.3: training only on synthetic AnomLLM, then testing on mixed synthetic-real VisualTimeAnomaly and real-world TSB-UAD. This shows transferrability to unseen datasets, domains, and sequence lengths.
> * We also evaluated on three dynamic real-world datasets: IOPS (high noise), ECG (dynamic frequency), and Yahoo (complex seasonality). Ours remain robust under severe non-stationarity (see below).
>
> | |GPT-4o|Gemini-2.5-Pro|Ours|
> |-|-|-|-|
> |IOPS|38.5|48.5|76.4|
> |ECG|21.2|48.4|81.0|
> |Yahoo|75.2|78.4|88.2|
>
> **Why does AnomSeer generalize?** Instead of optimizing only final outcomes, TimerPO aligns intermediate reasoning with expert statistical traces. By learning the logic of structural shifts rather than superficial cues, the model transfers to unseen, non-stationary settings.
>
> ### Q2 & W2: Detection of unseen anomaly patterns
> AnomSeer covers the standard TSAD taxonomy (e.g., point and range) and nearly all common anomaly types in the literature [1,2].
> * **Zero-shot on new types**. As shown in Fig. 6, AnomSeer successfully detects shapelet anomalies absent from training.
> * **Robustness to incomplete data**. We also evaluate extreme distortions with missing data. Although performance degrades as continuity is disrupted, AnomSeer remains clearly stronger than commercial APIs, with a clear advantage at 40% missing data, though the margin narrows at 60%.
>
> | |GPT-4o|Gemini-2.5-Pro|Ours|
> |-|-|-|-|
> |Miss 20%|49.1|52.4|75.1|
> |Miss 40%|45.8|48.2|68.9|
> |Miss 60%|37.4|42.5|45.8|
>
> So the detection boundary is mainly determined by how much temporal structure remains continuous. As long as enough interpretable visual context is preserved, the model can generalize to unseen patterns and distortions.
>
> [1] TAB: Unified Benchmarking of Time Series Anomaly Detection Methods. PVLDB'25
>
> [2] Can LLMs Understand Time Series Anomalies? ICLR'25
>
> ### W3: AnomLLM: how well does it reflect real-world anomaly diversity and complexity?
> AnomLLM covers common anomaly types in controlled settings, providing a simplified yet meaningful approximation of real-world anomaly diversity and complexity. Compared with real-world data (TSB-UAD), it has about 3.3× shorter median sequence length, 18.4× denser anomalies, and 7.5× fewer anomalous segments per series. We use it as a structured curriculum with high-fidelity supervision to cultivate robust reasoning, and Sec. 5.3 demonstrates our model’s superior generalization on diverse, complex real-world data.
>
> ### Q3: Does tokenizing long time series hurt accuracy, especially for local or between-token anomalies?
> **Not** significantly. Unlike numerical tokenization, which may split a local pattern into isolated symbols, AnomSeer uses visual line plots, where temporal structures stay spatially continuous. Although the MLLM backbone encodes the image into patch tokens, self-attention in Qwen2-VL [3] integrates information across patch boundaries and the full visual field. So anomalies spanning multiple patches are modeled jointly rather than within a single token, making the representation less sensitive to token boundaries and more effective for long-range dependencies and between-token anomalies.
>
> **Very local anomalies are further preserved through fine-grained statistical grounding**. ExpCoT uses a hierarchical scan with anomaly-specific probes to detect local discords. Thus, even visually small anomalies or those near patch boundaries can still be identified at the time-series level and reinforced by TimerPO. Tab. 1 shows strong performance on subtle anomalies and Tab. 7 further supports robustness on short-duration and boundary-sensitive cases.
>
> Compared with direct numerical tokenization, AnomSeer is less prone to missing highly local or boundary-crossing anomalies because it combines **spatially continuous visual representations, cross-patch attention, and fine-grained statistical grounding**.
>
> [3] Qwen2-VL: Enhancing vision-language model's perception of the world at any resolution. arXiv:2409.12191
>
> ### Q4: Performance under resource-constrained deployment
> We apply post-training quantization (PTQ) on our 3B model in an edge-style setup (2 CPU cores, 32GB RAM). The average F1 scores are reported below.
>
> | |Prec.|Mem.|F1|
> |-|-|-|-|
> |FP16|16/16-bit|6.5GB|79.3|
> |Shared-scale FP8|8/8-bit|~3.4GB|78.6|
> |Shared-exp FP4|4/8-bit|~2.1GB|76.2|
>
> F1 only slightly degrades under tighter resources: 8-bit PTQ reduces memory to ~3.4GB with only a 0.7% F1 drop, while 4-bit reduces it to ~2.1GB with a 3.1% drop. So even below the recommended threshold, ours remain practical, with 8-bit PTQ offering the best trade-off.
>
> ### Reply to ''Limitation''
> * AnomSeer naturally supports multivariate reasoning and remains robust in multi-channel settings (Appendix E.4).
> * We support low-cost inference on basic hardware like a 2-core CPU (Q4).

---

> > ### Author Rebuttal · Reviewer_bmp3 · 2026-04-04
> >
> > The rebuttal has addressed all the questions I raised earlier, but given that the final score is based on the original paper, I have decided to keep my previous rating.

---

> > > ### Author Response · Authors · 2026-04-06
> > >
> > > Thank you for your thoughtful feedback. We are glad that our rebuttal has successfully addressed all your concerns!
> > >
> > > Our original paper already covers Q1, Q2, and Q3, as discussed in Sec. 5.3 and Sec. E.4. As for dynamic datasets, more unseen anomaly types, and resource-constrained deployment, we will add them to the final version.
> > >
> > > Thank you again for your helpful comments.

---

### Official Review · Reviewer_8N18 · 2026-03-12

**Soundness:** 3
**Presentation:** 3
**Significance:** 3
**Originality:** 3
**Overall Recommendation:** 4
**Confidence:** 4

**Summary:**

This paper proposes a post training method for multi-modal LLMs on time-series anomaly detection with data generated by statistical methods.

**Compliance With Llm Reviewing Policy:**

Affirmed.

**Final Justification:**

The rebuttal addresses my concerns. I decided to maintain my original score. Good Luck.

**Key Questions For Authors:**

Please see weakness

**Limitations:**

Yes

**Strengths And Weaknesses:**

# Strength

S1: The idea of constructing expert trajectories from classical statistical tools is reasonable and seems work well backed by empirical result.

S2: The ablation study is generally informative. Overall, the experiment section is fairly solid and supports the motivation of the work.

# Weakness / Questions

WQ1: The baseline evaluated is restricted to LLM/MLLM-based methods. I think the paper would be stronger if it also compared against traditional TSAD methods such as ones used in [1] and time series anomaly-detection ML baselines such as [2,3]. Although these methods do not generate reasoning traces, practical usefulness still depends on detection performance.

WQ2: ExpCoT is designed with a three stage structure. This staged design is quite intuitive, but it also have a fairly strong inductive bias. The current ablations show that ExpCoT as a whole helps, but it does not quite disentangle whether the gains come from the staged structure itself, or from the statistical evidence used in the trace. A more targeted study would make the role of ExpCoT clearer. For example, the authors could sample alternative reasoning trajectories with the same underlying statistical evidence but different organizations or levels of detail, and test whether the staged format itself is important.

WQ3: More of a follow up to WQ2. ExpCoT contains structured reasoning and the correct anomaly type with localization. This makes it somewhat unclear whether the gain comes from reasoning supervision or simply from label supervision. A useful control study would be to preserve the reasoning format while perturbing the final label or localization to test whether the reasoning structure itself provides learning value beyond just carrying the correct solution.

[1]: TSB-UAD: An End-to-End Benchmark Suite for Univariate Time-Series Anomaly Detection

[2]: TimesNet: Temporal 2D-Variation Modeling for General Time Series Analysis

[3]: MOMENT: A Family of Open Time-series Foundation Models

---

> ### Author Rebuttal · Authors · 2026-03-31
>
> We thank you for the constructive feedback and are encouraged that the core ideas and experiments were found supportive of the paper’s motivation.  We address the comments point by point below.
>
> ### W1: Comparison with traditional TSAD and ML baselines
>
> Thank you for the helpful feedback. We added the comparisons with suggested TSAD methods: those used in [1] and ML baselines [2,3].
>
> | **Model** | **Freq.** | **Trend** | **Range** | **Point** | **Avg** |
> | --- | --- | --- | --- | --- | --- |
> | SR [4] | 65.9 | 18.0 | 28.5 | 27.8 | 35.1 |
> | LSTi [5] | 11.4 | 29.4 | 67.2 | 87.4 | 48.9 |
> | DWT [6] | 57.1 | 58.5 | 55.8 | 65.9 | 59.3 |
> | TimesNet [2] | 69.5 | 85.4 | 81.2 | 77.5 | 78.4 |
> | MOMENT [3] | 71.2 | 76.8 | 75.9 | 75.3 | 74.8 |
> | Ours | 60.8 | 87.7 | 94.3 | 94.9 | 84.4 |
>
> These results show that our model achieves the best overall results on TSAD. Traditional TSAD methods can be competitive on some specific anomaly types, especially frequency-related ones, but are less consistent across anomaly families. TimesNet is strong on long-term trends and interval anomalies, and MOMENT is more balanced in zero-shot settings. In contrast, AnomSeer achieves **stronger overall detection performance while also providing reasoning traces**.
>
> [1]: TSB-UAD: An End-to-End Benchmark Suite for Univariate Time-Series Anomaly Detection
>
> [2]: TimesNet: Temporal 2D-Variation Modeling for General Time Series Analysis
>
> [3]: MOMENT: A Family of Open Time-series Foundation Models
>
> [4] Time-series anomaly detection service at Microsoft. KDD 2019
>
> [5] Matrix Profile I: all pairs similarity joins for time series: a unifying view that includes motifs, discords and shapelets. ICDM 2016
>
> [6] Time series anomaly detection with discrete wavelet transforms and maximum likelihood estimation. ITISE 2017
>
> ### W2: Staged structure vs. statistical evidence in ExpCoT
>
> We agree that a reordered-trace control would be valuable. We conduct a more targeted study by varying the organization and the level of detail of the reasoning trajectory while keeping the rest of training unchanged.
>
> | **Variant** | **Freq** | **Trend** | **Range** | **Point** | **Avg** |
> | --- | --- | --- | --- | --- | --- |
> | ExpCoT (Full) | 58.9 | 84.9 | 85.6 | 87.8 | 79.3 |
> | w/o staged structure | 55.2 | 81.5 | 82.5 | 84.1 | 75.8 (↓3.5) |
> | w/o precise evidence | 51.4 | 79.1 | 82.3 | 81.7 | 73.6 (↓5.7) |
>
> When we keep the same underlying statistical evidence but shuffle the three-stage structure, performance drops by 3.5%. This suggests that the **staged format** itself matters. It serves as an effective scaffold, organizing the evidence into a coherent reasoning trajectory while **preserving alignment between the reasoning process and the final conclusion**.
>
> When we preserve the staged structure but reduce the level of detail of the statistical evidence, replacing precise values (e.g., z-scores or FFT values) with qualitative descriptions such as “significant deviation,” performance drops more sharply by 5.7%. This shows that **the statistical evidence appears to contribute more strongly to the gain**.
>
> Overall, these results suggest that **both factors matter**: the statistical evidence provides the main signal, while the staged structure contributes additional value by organizing that evidence into a consistent reasoning trajectory.
>
> ### W3: Reasoning supervision vs. label supervision
>
> Thank you for your valuable suggestions. As suggested, we conduct a control study that preserves the reasoning format while perturbing the final label/localization in the Conclusion stage.
>
> | **Training supervision** | **Final answer** | **Reasoning structure** | **OOD (Shapelet) F1** | **In-domain F1** |
> | --- | --- | --- | --- | --- |
> | AnomSeer (Full) | Correct | Correct | 77.5 | 79.3 |
> | Perturbed label | Incorrect | Correct | 75.7 | 77.4 |
> | Label-only  | Correct | N/A | 70.8 | 72.8 |
>
> The result is clear: even when the final label/localization is perturbed, the model still substantially outperforms the label-only baseline, especially on OOD Shapelet detection. This indicates that **the gains do not come only from carrying the correct solution in the trace**. The reasoning structure itself provides **useful learning signal beyond simple label supervision**.
>
> At the same time, the perturbed-label model remains below the full model, showing that **the best performance requires both correct reasoning supervision and correct final supervision**. Overall, this control study supports the claim that **ExpCoT provides learning value beyond simply encoding the correct anomaly type and localization**.

---

> > ### Author Rebuttal · Reviewer_8N18 · 2026-04-03
> >
> > The rebuttal addresses my concerns. I decided to maintain my original score. Good Luck.

---

> > > ### Author Response · Authors · 2026-04-06
> > >
> > > Thank you for your positive feedback. We are pleased that our rebuttal has addressed your concerns. We sincerely appreciate your time and effort in reviewing our paper.

---

### Official Review · Reviewer_NV2A · 2026-03-13

**Soundness:** 3
**Presentation:** 3
**Significance:** 3
**Originality:** 3
**Overall Recommendation:** 4
**Confidence:** 4

**Summary:**

This paper presents AnomSeer, a framework designed to improve Time-series anomaly detection (TSAD) with multimodal large language models (MLLMs) by shifting reliance from visual cues to quantitative reasoning. The method combines ExpCoT, which generates reasoning traces using classical statistical and frequency analyses, with TimerPO, a reinforcement learning approach that utilizes an Optimal Transport-based advantage signal and orthogonal projection. Trained on synthetic data, AnomSeer demonstrates improved accuracy compared to larger baselines on both synthetic and real-world benchmarks, while producing structured reasoning traces for anomaly explanation.

**Compliance With Llm Reviewing Policy:**

Affirmed.

**Final Justification:**

While the original submission had clarity gaps regarding ExpCoT examples and multivariate support, the authors fully addressed these concerns in the rebuttal with supplementary experiments and analysis. Given that the rebuttal resolved the key weaknesses and reinforced the method's originality and soundness, I raised my the Overall Recommendation from 3 to 4.

**Key Questions For Authors:**

- Could specific examples be provided to illustrate how TimerPO facilitates the detection of fine-grained patterns more effectively than GRPO?


- Could additional results or reasoning be provided to demonstrate that the synthetic data generation method (ExpCoT) possesses robust generalization capabilities?


- Could further analysis be provided, specifically regarding whether the method can be extended to support multivariate time series and whether it offers performance advantages over statistical approaches?

**Limitations:**

yes

**Strengths And Weaknesses:**

Strengths:


- This paper introduces the ExpCoT method, which utilizes structured templates to guide the generation of chain-of-thought traces for TSAD. Compared to unguided chain-of-thought generation, this approach enhances the model's capability in analyzing time-series data better.


- This paper introduces the TimerPO method, which leverages Optimal Transport to provide fine-grained rewards for reasoning traces. Compared to using outcome-only rewards, this approach facilitates the analysis of subtle patterns and discrepancies.


- Evaluation results indicate that the framework achieves moderate performance improvements compared to prior state-of-the-art baselines. The model demonstrates consistent performance across different datasets, reflecting its generalization capability.


Weaknesses:


- Writing. The specific construction methodology and illustrative examples of ExpCoT are not provided in the main text. Furthermore, critical baseline comparisons (e.g., conventional TSAD detectors) and the support for multivariate time series have not been demonstrated in the main content.


- There is a lack of explicit alignment between the statistical evidence that ExpCoT relies upon and the model's visual input modalities, and the visual patterns have not been demonstrated to correspond to the statistics within ExpCoT.


- Compared to the main results, the ablation studies omit performance metrics for the classification task. Furthermore, while different scenarios exhibit varying degrees of sensitivity to the ablation of specific components, an in-depth analysis of these variations is not provided.

---

> ### Author Rebuttal · Authors · 2026-03-31
>
> Thanks for your thoughtful comments. We address them as follows.
> ### W1: Writing on ExpCoT, baselines, multivariate support
> Thank you for the helpful suggestion. While we have provided illustrative ExpCoT, baseline comparisons, and multivariate support **in the appendix C.2**, we agree they should be more visible in the main paper. In the revision, we will:
> * Move a concrete ExpCoT and one example (currently Appendix C.1/C.2) to Sec. 4.1 (ExpCoT Generation), after the three-stage ExpCoT description;
> * Move the comparison with conventional TSAD baselines (currently Appendix E.5) to Sec. 5.1, after the main benchmark table, so comparisons with both MLLM-based and traditional TSAD methods are visible in the main text;
> * Move the discussion and results on multivariate support (currently Appendix E.4) to Sec. 5.3 (Generalization Performance), with a corresponding clarification in Sec. 6 (Limitations).
> ### W2: Alignment of visual and statistical evidence
> This alignment is inherent in ExpCoT. In Sec. 4.1, Observation identifies visual patterns, and Reasoning links them to anomaly-specific tests, e.g., spikes to Matrix Profile and slope changes to gradient analysis. The resulting statistics are then converted into timestamp-specific natural-language evidence.
>
> Fig. 5 supports this: before TimerPO, outputs are narrower and less aligned with ExpCoT, dominated by coarse terms like "global" and "change"; after TimerPO, the vocabulary shifts to grounded temporal terms such as "timestamp" and "amplitude", indicating more grounded reasoning.
> ### W3: Classification metrics and sensitivity
> Thank you for the helpful suggestion. We add classification ACC to complement Tab 2:
> |  | Freq. | Tre. | Ran. | Poi. |
> | - | -| - | - | - |
> | w/o ExpCoT | 43.8 | 58.9 | 56.3 | 50.1 |
> | w/o Orth. | 48.1 | 66.0 | 60.4 | 54.3 |
> | GRPO | 46.2 | 61.5 | 58.1 | 51.8 |
> | Ours | 54.7 | 71.6 | 65.1 | 59.8 |
>
> The results show clear sensitivity: freq anomalies depend most on reasoning guidance, trend anomalies show the largest classification drop without ExpCoT, and removing orthogonal integration consistently hurts all categories. So ExpCoT improves anomaly-type disambiguation and structural grounding, orthogonal integration stabilizes reasoning supervision, and the full TimerPO design is especially beneficial for subtle patterns (e.g., frequency drift).
> ### Q1: Give examples showing that TimerPO performs better than GRPO for fine-grained patterns.
> We have supported this with distribution-, case-, and performance-level evidence.
> * Fig 12 shows that GRPO is dominated by coarse terms such as “*expected*”, while TimerPO shifts toward finer temporal terms such as “*intervals”*, and “*amplitude”*.
> * Fig 3 gives a concrete case: our model recognizes the repeating waveform, finds a local disruption around timestamp 149, and localizes the contextual anomaly to [140,155], while the GRPO-trained model follows a global visual heuristic and misses the local structural break.
> *  Tab. 2 shows the same trend quantitatively, especially for frequency anomalies, as seen in GRPO (row 3) vs. our method (row 4). The main reason is that GRPO optimizes only globally verifiable outcomes, while TimerPO further aligns learning with fine-grained expert reasoning via the time-series reasoning advantage and orthogonal integration.
> ### Q2: Generalization of ExpCoT
> * Empirically, Fig. 6 shows zero-shot transfer from synthetic training data to real-world data without adaptation, and successful detection of unseen shapelet anomalies, supporting transfer across datasets and anomaly types.
> * Theoretically, ExpCoT + TimerPO adds a process-level constraint beyond outcome-only RL, restricting learning from a broad policy class to a subset $H_\delta \subseteq H$ whose reasoning traces stay within deviation $\delta$ of expert traces. Under a standard capacity-based view [1][2],
>
> $$
> R(h)\le \hat{R}(h)+2R_n(H_\delta)+O\left(\sqrt{\tfrac{\log(1/\eta)}{n}}\right)+L\delta,
> $$
>
> As $H_\delta \subseteq H$, we have $R_n(H_\delta)\le R_n(H)$. So ExpCoT acts as a structured regularizer, favoring stable reasoning patterns grounded in domain-agnostic signals such as FFT, gradient, and Matrix Profile, which transfer better across domains.
>
> [1] Rademacher and Gaussian Complexities: Risk Bounds and Structural Results. JMLR'02.
>
> [2] A Vector-Contraction Inequality for Rademacher Complexities. ALT'16.
> ### Q3: Support for Multivariate time series
> * We have provided this support evidence in Appendix E.4, where we extended AnomSeer to multivariate inputs via subplot rendering. As shown in Appendix E.4, AnomSeer outperformed frontier models (e.g., GPT-4o Gemini-2.5).
> * Here, we add comparisons with statistical approaches. AnomSeer offers a clear performance advantage:
>
> | | Cap.  | Syn. | Real |
> | - | - |-| - |
> | FFT | Loc. | 68.4 | 51.6 |
> | MP | Loc.| 55.1 | 66.5 |
> | Grad.| Loc.| 59.4 | 55.7 |
> | Ours | Loc.+Cls.+Rea. | 83.5 | 72.4 |

---

> > ### Author Rebuttal · Reviewer_NV2A · 2026-04-04
> >
> > My concerns have been adequately addressed. Based on the promising supplementary experiments and analyses presented in the authors' rebuttal, I raised my Overall Recommendation from 3 to 4.

---

> > > ### Author Response · Authors · 2026-04-06
> > >
> > > Thank you for raising the overall score. We are glad that our rebuttal has addressed your concerns. We also sincerely appreciate your time and feedback, which have helped improve the quality of our work.

---

### Decision · Program_Chairs · 2026-04-30

**Decision:**

Accept (regular)

**Comment:**

This paper presents a timely and well-motivated effort to improve time-series anomaly detection with MLLMs by shifting from superficial visual pattern matching toward more structured quantitative reasoning. The combination of ExpCoT and TimerPO to be meaningful and technically sound. The empirical results, together with the added rebuttal experiments, provide reasonably evidence of effectiveness, robustness, and generalization across datasets and different settings. While some presentation and analysis issues remain, especially regarding exposition of ExpCoT construction and alignment between visual inputs and statistical evidence, these concerns appear addressable and do not outweigh the paper’s overall strengths. Overall, the paper offers a promising and practically relevant contribution for time series analysis, and I recommend acceptance.